 

# Parathyroid hormone attenuates osteoarthritis pain by remodeling subchondral bone in mice

Qi Sun[1,2], Gehua Zhen[1], Tuo Peter Li[1], Qiaoyue Guo[1], Yusheng Li[1], Weiping Su[1], Peng Xue[1], Xiao Wang[1], Mei Wan[1], Yun Guan[3], Xinzhong Dong[4,5,6,7], Shaohua Li[2], Ming Cai[2], Xu Cao[1,8]*

[1]Department of Orthopaedic Surgery, Institute of Cell Engineering, The Johns Hopkins University School of Medicine, Baltimore, United States; [2]Department of Orthopaedics, Shanghai Tenth People's Hospital, School of Medicine, Tongji University, Shanghai, China; [3]Department of Anesthesiology and Critical Care Medicine, The Johns Hopkins University School of Medicine, Baltimore, United States; [4]Department of Neuroscience, The Johns Hopkins University School of Medicine, Howard Hughes Medical Institute, Baltimore, United States; [5]Department of Neurosurgery, The Johns Hopkins University School of Medicine, Howard Hughes Medical Institute, Baltimore, United States; [6]Department of Dermatology, The Johns Hopkins University School of Medicine, Howard Hughes Medical Institute, Baltimore, United States; [7]Center of Sensory Biology, The Johns Hopkins University School of Medicine, Howard Hughes Medical Institute, Baltimore, United States; [8]Department of Biomedical Engineering, The Johns Hopkins University School of Medicine, Baltimore, United States

*For correspondence:
xcao11@jhmi.edu

Competing interests: The authors declare that no competing interests exist.

**Abstract** Osteoarthritis, a highly prevalent degenerative joint disorder, is characterized by joint pain and disability. Available treatments fail to modify osteoarthritis progression and decrease joint pain effectively. Here, we show that intermittent parathyroid hormone (iPTH) attenuates osteoarthritis pain by inhibiting subchondral sensory innervation, subchondral bone deterioration, and articular cartilage degeneration in a destabilized medial meniscus (DMM) mouse model. We found that subchondral sensory innervation for osteoarthritis pain was significantly decreased in PTH-treated DMM mice compared with vehicle-treated DMM mice. In parallel, deterioration of subchondral bone microarchitecture in DMM mice was attenuated by iPTH treatment. Increased level of prostaglandin E2 in subchondral bone of DMM mice was reduced by iPTH treatment. Furthermore, uncoupled subchondral bone remodeling caused by increased transforming growth factor β signaling was regulated by PTH-induced endocytosis of the PTH type 1 receptor–transforming growth factor β type 2 receptor complex. Notably, iPTH improved subchondral bone microarchitecture and decreased level of prostaglandin E2 and sensory innervation of subchondral bone in DMM mice by acting specifically through PTH type 1 receptor in Nestin[+] mesenchymal stromal cells. Thus, iPTH could be a potential disease-modifying therapy for osteoarthritis.

## Introduction

Osteoarthritis is the most common degenerative joint disorder in the USA and a leading cause of disability (*Centers for Disease Control and Prevention (CDC), 2009*; *Murray et al., 2012*). Chronic pain is a prominent symptom of osteoarthritis, affecting nearly 40 million people in the USA (*Peat et al., 2001*). Pain itself is also a major risk factor for the development of functional limitation

**eLife digest** Over time the cartilage between our bones gets worn down, and this can lead to a painful joint disorder known as osteoarthritis. Nearly 40 million people with osteoarthritis in the United States experience chronic pain. Although there are a number of drugs available for these patients, none of them provide sustained pain relief, and some have substantial side effects when ingested over a long period of time.

Bone tissue is continuously broken down into minerals, such as calcium, that can be reabsorbed into the blood. In 2013, a group of researchers found that the tissue in the layer of bone below the cartilage – known as the subchondral bone – is reabsorbed and replaced incorrectly in patients with osteoarthritis. This irregular 'remodeling' stimulates nerve cells to grow into the subchondral layer, leading to increased sensitivity in the joint.

A protein called parathyroid hormone, or PTH for short, plays an important role in the loss and formation of bone. A drug containing PTH is used to treat patients with another bone condition called osteoporosis, and could potentially work as a treatment for osteoarthritis pain.

To investigate this, Sun et al. – including some of the researchers involved in the 2013 study – tested this drug on a mouse model that mimics the symptoms of osteoarthritis. This revealed that PTH significantly decreases the number of nerves present in the subchondral bone, which caused the mice to experience less pain. PTH also slowed down the progression of osteoarthritis, by preventing the cartilage on the subchondral layer from deteriorating as quickly. Sun et al. found that the subchondral bones of treated mice also had a more stable structure and reduced levels of a protein involved in the reabsorption of bone.

The results suggest that PTH is able to correct the errors in bone remodeling caused by osteoarthritis, and that this drug could potentially alleviate patients' chronic pain. This drug has already been approved by the US Food and Drug Administration (FDA), and could be used in clinical trials to see if PTH has the same beneficial effects on patients with osteoarthritis.

and disability in patients with osteoarthritis (*Lane et al., 2010*). Unfortunately, treating osteoarthritis pain is challenging and represents a substantial unmet medical need. Available therapies (nonsteroidal anti-inflammatory drugs, analgesics, and steroids) do not provide sustained pain relief and can have substantial adverse effects (*Geba et al., 2002*; *Karlsson et al., 2002*). Inadequate control of chronic osteoarthritis pain is a major reason that patients seek surgical treatment.

The sources and mechanisms of osteoarthritis pain may be multifactorial. Some studies have focused on synovial inflammation and cartilage degeneration (*Mathiessen and Conaghan, 2017*; *Malfait and Schnitzer, 2013*). Yet, osteoarthritis pain can manifest during very early stages of the disease, in the absence of synovial inflammation and independently of progressive cartilage degeneration. Many patients have radiographic osteoarthritic changes but no symptoms, whereas others have osteoarthritis pain with no radiographic indications (*Dieppe and Lohmander, 2005*; *Hannan et al., 2000*; *Bedson and Croft, 2008*). Some patients have bilateral radiographic evidence of knee osteoarthritis yet have unilateral knee pain. Little attention has been paid to subchondral bone to control osteoarthritis pain. Intriguingly, subchondral bone marrow edema–like lesions are highly correlated with osteoarthritis pain (*Yusuf et al., 2011*; *Kwoh, 2013*). Zoledronic acid, which inhibits osteoclast activity, reduces the size of bone marrow edema–like lesions and concomitantly alleviates pain (*Laslett et al., 2012*). Analysis of data from the National Institutes of Health Osteoarthritis Initiative reported that patients taking bisphosphonates experienced significantly reduced knee pain at years 2 and 3 (*Laslett et al., 2014*). Furthermore, patients with osteoarthritis report rapid and obvious pain relief after removal of deteriorated subchondral bone with overlying cartilage through knee replacement (*Isaac et al., 2005*; *Reilly et al., 2005*). Articular cartilage, which is characteristically aneural and avascular (*Bhosale and Richardson, 2008*), is incapable of generating pain primarily, suggesting that subchondral bone is a major source of osteoarthritis pain.

Aberrant subchondral bone remodeling is a critical factor in pathological changes of osteoarthritis (*Zhen and Cao, 2014*; *Zhen et al., 2013*), and sensory innervation induced by netrin-1 from aberrant subchondral bone remodeling is responsible for osteoarthritis pain (*Zhu et al., 2019*). Furthermore, locally elevated levels of prostaglandin E2 (PGE2) activate sensory nerves in porous endplates,

leading to sodium influx through Na$_v$1.8 channels, to mediate spinal hypersensitivity (*Ni et al., 2019*). We have shown that bone homeostasis is maintained by temporal-spatial activation of transforming growth factor β (TGF-β) to couple bone resorption and formation (*Tang et al., 2009*), in which subchondral bone is maintained in a native microarchitecture with blood vessels and nerves intertwined under normal conditions. However, excessive active TGF-β in the subchondral bone induces aberrant bone remodeling at the onset of osteoarthritis to promote its progression (*Zhen et al., 2013*). Specifically, high levels of active TGF-β recruit mesenchymal stromal cells (MSCs) and osteoprogenitors in clusters, leading to abnormal bone formation and angiogenesis (*Zhen et al., 2013*).

Parathyroid hormone (PTH), a U.S. Food and Drug Administration–approved anabolic agent for osteoporosis, regulates bone remodeling and calcium metabolism (*Crane and Cao, 2014*; *Qiu et al., 2010*). The parathyroid gland, the main production site of PTH, evolved in amphibians and supported the transition from aquatic to terrestrial life, allowing terrestrial locomotion in previously aquatic vertebrates (*Okabe and Graham, 2004*). PTH prevents cartilage degeneration and/or the deterioration of subchondral bone (*Sampson et al., 2011*; *Orth et al., 2013*; *Bellido et al., 2011*; *Yan et al., 2014*; *Chang et al., 2009*; *Lugo et al., 2012*; *Eswaramoorthy et al., 2012*; *Cui et al., 2020*; *Morita et al., 2018*; *Dai et al., 2016*; *Chen et al., 2018*), induces cartilage regeneration or chondrocytes proliferation in osteoarthritis (*Sampson et al., 2011*; *Petersson et al., 2006*), and stimulates subchondral bone and articular cartilage repair of focal osteochondral defects (*Orth et al., 2013*). PTH also interacts with local osteotropic factors to orchestrate the coupling of bone resorption and formation in an anabolic signaling network (*Qiu et al., 2010*; *Pfeilschifter et al., 1995*), and PTH and TGF-β work in concert to exert their physiological activities in bone (*Qiu et al., 2010*). TGF-β elicits its cellular response through the ligand-induced formation of a heteromeric complex containing TGF-β types 1 (TβRI) and 2 (TβRII) kinase receptors (*Tang et al., 2009*; *Wrana et al., 1992*). We have shown that PTH induces endocytosis of PTH type one receptor (PTH1R) with TβRII as a complex, and both PTH and TGF-β signaling are co-regulated during endocytosis (*Qiu et al., 2010*). In this study, we investigated whether intermittent parathyroid hormone (iPTH) could attenuate osteoarthritis pain by modifying cartilage degeneration, subchondral bone microarchitecture, and subchondral sensory innervation. We found that iPTH reduces osteoarthritis pain and attenuates progression of osteoarthritis by attenuating subchondral bone deterioration and cartilage degeneration. Furthermore, iPTH reduces sensory innervation and the level of PGE2 in the subchondral bone and improves subchondral bone remodeling specifically through PTH1R on Nestin[+] MSCs.

## Results

### iPTH attenuates osteoarthritis pain and articular cartilage degeneration

To examine the effect of iPTH on osteoarthritis pain, we generated a DMM model of osteoarthritis in 10 week old mice, administered daily PTH subcutaneously beginning 3 days after surgery, and assessed pain behavior during the following 8 weeks. We assessed three types of pain behavior: secondary hyperalgesia, primary hyperalgesia, and gait deficit. Secondary hyperalgesia of the affected hind paw was analyzed weekly by measuring 50% paw withdrawal threshold (50% PWT). During the first 2 weeks after surgery, 50% PWT decreased uniformly in all three groups of mice: sham-operated mice ('sham mice'), DMM mice that received vehicle daily ('vehicle mice'), and DMM mice that received PTH daily ('PTH mice') (*Figure 1A*). The 50% PWT of sham mice returned to baseline by week 3 after surgery, indicating that secondary hyperalgesia during the first 2 weeks is attributable to surgery alone. The 50% PWT of vehicle mice continued to worsen from week 2 onward, but the 50% PWT of PTH mice stabilized at levels significantly higher than those of vehicle mice (*Figure 1A*), indicating that iPTH initiated early in the development of osteoarthritis reduced secondary hyperalgesia.

Primary knee hyperalgesia was analyzed weekly by measuring withdrawal threshold during direct knee press using a pressure application measurement (PAM) force transducer as previously described (*Miller et al., 2017*). At week 1 after surgery, PAM withdrawal threshold (PAMWT) decreased uniformly in all three groups (*Figure 1B*). PAMWT of sham mice recovered to near-baseline by week 4. PAM withdrawal threshold of vehicle mice also improved by week 4, albeit to a lesser

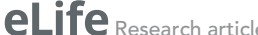

**Figure 1.** Parathyroid hormone (PTH) improves osteoarthritis pain and joint degeneration after DMM surgery. (**A**) 50% paw withdrawal threshold (50% PWT) was tested in the left hind paw of sham-operated, PTH-treated, destabilized medial meniscus (DMM), and vehicle-treated DMM mice at different time points (n = 8/group). [#, ##]Vehicle-treated DMM mice compared with sham mice; [*,**]Vehicle-treated DMM mice compared with PTH-treated DMM mice. (**B**) Withdrawal threshold measured by pain application measurement (PAMWT) at the left knee of sham-operated, PTH-treated DMM, and vehicle-treated DMM mice (n = 8/group). [#, ##]Vehicle-treated DMM mice compared with sham mice; [*,**]Vehicle-treated DMM mice compared with PTH-treated DMM mice. (**C**) Representative images of gait analysis of sham-operated, PTH-treated DMM. and vehicle-treated DMM mice. RH = right hind paw (pink), LH = left hind paw (green), RF = right front paw (blue), LF = left front paw (yellow). (**D**) Quantitative analysis of LH intensity, LH area, and LH swing speed compared with the RH at week 8 after sham or DMM surgery (n = 8/group). (**E**) Safranin O/fast green (SOFG) staining of sagittal sections of the tibial medial compartment, proteoglycan (red) and bone (green) at week 8 after sham or DMM surgery. Age of mice used for sections: 18 weeks old. Scale bar: 250 μm. (**F**) Osteoarthritis Research Society International (OARSI) scores at weeks 2, 4, and 8 after surgery (n = 8/group). (**G**) Immunohistochemical analysis of matrix metalloproteinase 13+ (MMP13+, brown) and type X collagen+ (ColX+, brown) in articular cartilage at week 8 after sham or DMM surgery. Age of mice used for sections: 18 weeks old. Scale bar: 50 μm. (**H**) Quantitative analysis of MMP13+ and ColX+ cells in articular cartilage. All data are shown as means ± standard deviations (n = 8/group). [*,#]p<0.05, [**,##]p<0.01. NS, no significant difference.
*Figure 1 continued on next page*

*Figure 1 continued*

The online version of this article includes the following source data for figure 1:

**Source data 1.** Raw data of PAMWT, 50% PWT, catwalk analysis, OARSI, MMP13$^+$ staining, and ColX$^+$ staining.

extent compared with sham mice, and gradually declined between weeks 4 and 8. PAMWT of PTH mice and vehicle mice changed in a similar pattern over time; however, the PAMWT of PTH mice was significantly higher than that of vehicle mice at every time point between week 2 and week 8 after surgery (*Figure 1B*), indicating that iPTH initiated early in the development of osteoarthritis reduced primary hyperalgesia.

Gait deficit was analyzed by performing gait analysis biweekly, comparing the swing speed, contact area, and paw intensity between the affected and contralateral hind paws (*Figure 1C*). At week 8 after DMM surgery (compared with sham surgery), the affected hind paw intensity, contact area, and swing speed decreased significantly relative to contralateral hind paws; however, these differences were less pronounced in PTH mice (*Figure 1D*), indicating that iPTH initiated early in the development of osteoarthritis improved gait deficits.

To examine the effect of iPTH on degeneration of articular cartilage in osteoarthritis, we performed Safranin O/fast green (SOFG) staining (*Figure 1E*) and used the Osteoarthritis Research Society International (OARSI) histologic grading system to assess articular cartilage damage. OARSI score increased progressively in DMM mice over 8 weeks, but this progression was reduced and slower in PTH mice (*Figure 1F*), indicating that early initiation of iPTH in osteoarthritis slowed the progression of articular cartilage degeneration. We also assessed the effect of iPTH on chondrocyte degeneration by performing immunohistochemical analysis to measure the percentage of type X collagen-containing (ColX$^+$) and matrix metalloproteinase 13-containing (MMP13$^+$) chondrocytes in articular cartilage (*Figure 1G*). Percentages of ColX$^+$ and MMP13$^+$ chondrocytes more than doubled in DMM mice compared with sham mice, and iPTH significantly reduced the magnitude of increase (*Figure 1H*), indicating that iPTH inhibited chondrocyte degeneration in osteoarthritis. These results indicate that iPTH initiated early in osteoarthritis reduced pain and attenuated cartilage degeneration.

## iPTH reduced nociceptive innervation and PGE2 level of osteoarthritic subchondral bone

To determine how iPTH reduced osteoarthritis pain, we examined the effect of iPTH on nociceptive nerve innervation in subchondral bone. DMM osteoarthritis joints were harvested at different time points for immunohistologic analysis of subchondral bone. The tibial subchondral bone was immunostained for calcitonin gene-related peptide (CGRP), the markers of peptidergic nociceptive C nerve fibers. The density of CGRP$^+$ nerve endings increased significantly after DMM surgery compared with sham surgery (*Figure 2A,B*). Although the density of CGRP$^+$ sensory nerve fibers also increased in PTH mice, the effect was significantly diminished relative to vehicle mice, indicating that iPTH inhibited the growth of nociceptive nerve fibers in osteoarthritic subchondral bone. Based on a newly proposed classification of sensory neurons (*Usoskin et al., 2015*), additional markers of nociceptive neurons including Substance P (SP) (*Figure 2C*), P2 × 3 and PIEZO2 (*Figure 2—figure supplement 1A,B*) were analyzed using immunofluorescence staining. The density of SP$^+$, P2 × 3$^+$, and PIEZO2$^+$ nociceptive fibers also increased in the subchondral bone of vehicle mice compared with sham mice, whereas iPTH treatment attenuated the increase of these nociceptive innervations (*Figure 2D*, *Figure 2—figure supplement 1C,D*). Sensory nerve fibers mostly innervate tissues together with small blood vessels. We also performed co-staining of CGRP and edomucin (EMUN) to examine the relationship between the CGRP$^+$ nerve fibers and EMUN$^+$ micro-vessels for investigating the effect of iPTH on the coupling of angiogenesis and sensory innervation in the subchondral bone. Like CGRP$^+$ nerve fibers, the density of EMUN $^+$ vessels increased in subchondral bone of vehicle mice compared with sham mice, while iPTH treatment attenuated the increase in these vessels (*Figure 2E,F*).

We immunostained for CGRP in the joint synovium to determine whether the effect of PTH on sensory innervation was specific to subchondral bone. The density of CGRP$^+$ sensory nerve endings increased after DMM surgery compared with sham surgery, and iPTH treatment had no influence on



**Figure 2.** Sensory nerve innervation in subchondral bone decreased with PTH treatment. (**A**) Immunofluorescence analysis of calcitonin gene-related peptide+ (CGRP+) (green) sensory nerve fibers in subchondral bone of sham-operated, PTH-treated DMM, and vehicle-treated DMM mice at week 8 after surgery. DAPI stains nuclei blue. Upper: low-magnification images (orange dotted line outlines the contour of the tibial subchondral bone), scale bar: 300 μm; bottom: high-magnification images. Scale bar: 50 μm. (**B**) Quantitative analysis of the density of CGRP+ sensory nerve fibers in tibial

*Figure 2 continued on next page*

Figure 2 continued

subchondral bone at week 8 after sham or DMM surgery (n = 8/group). (C, D) Immunofluorescence and quantitative analysis of substance P⁺ (SP⁺) (red) nerve fibers in subchondral bone at week 8 after sham or DMM surgery (n = 8/group). DAPI stains nuclei blue. Scale bar: 50 μm. (E, F) Representative images of immunofluorescence co-staining and quantitative analysis of the CGRP⁺ sensory nerves (green) and endomucin⁺ (EMUN) vessel (red) in the subchondral bone at week 8 after sham or DMM surgery (n = 8/group). Scale bar: 50 μm. (G, H) Immunofluorescence and quantitative analysis of CGRP⁺ (green) nerve fibers in synovium at week 8 after sham or DMM surgery. DAPI stains nuclei blue. Scale bar: 50 μm. n = 8/group. (I) μCT images of the tibial subchondral bone medial compartment (coronal) at week 8 after sham or DMM surgery. Arrowhead indicates the osteophyte. Scale bar: 500 μm. (J) Total volume measurement of osteophytes from the tibial plateau of sham-operated, PTH-treated DMM, and vehicle-treated DMM mice. (n = 8/group). (K) Immunohistochemical analysis of cyclooxygenase 2⁺ (COX2⁺) cells in the tibial subchondral bone at week 4 after DMM surgery. Arrowhead indicates the positive cells. Scale bar: 50 μm. (L) Quantitative analysis of COX2⁺ cells in mouse tibial subchondral bone (both bone marrow and subchondral bone matrix) (n = 8/group). (M) Quantitative analysis of PGE2 in subchondral bone determined by enzyme-linked immunosorbent assay (ELISA) (n = 8/group). *p<0.05, **p<0.01. NS, no significant difference.

The online version of this article includes the following source data and figure supplement(s) for figure 2:

**Source data 1.** Raw data of CGRP staining, SP staining, EMUN staining, CGRP staining in synovium, quantification osteophyte volume, COX2 staining, and quantification of level of PGE2.

**Figure supplement 1.** PTH treatment significantly reduced CGRP-positive nerve fibers in subchondral bone.

**Figure supplement 1—source data 1.** Raw data of P2 × 3 staining and PIEZO2 staining.

this effect (*Figure 2G,H*). Together, these findings suggest that iPTH reduces osteoarthritis pain by specific inhibition of sensory nerve innervation in subchondral bone.

Given that osteophytes can be responsible for increased pain sensation and abnormal gait behavior, we performed additional microcomputed tomography (μCT) analysis specifically examining the size of osteophyte in the DMM mice. There was no significant difference in osteophyte volume between PTH and vehicle-treated DMM mice, despite both are significantly higher than sham group (*Figure 2I,J*).

To examine the effect of iPTH on the PGE2 level in subchondral bone, we measured the expression level of cyclooxygenase 2 (COX2) by immunohistochemical analysis and measured the concentration of PGE2 by enzyme-linked immunosorbent assay (ELISA) in tibial subchondral bone. The number of COX2⁺ cell increased significantly after DMM surgery in vehicle mice compared with sham mice, and iPTH diminished this increase (*Figure 2K,L*). The level of PGE2 also increased after DMM surgery in vehicle mice compared with sham mice, and iPTH inhibited this effect (*Figure 2M*). These results indicate that early initiation of iPTH reduced the level of PGE2 in osteoarthritis subchondral bone.

## iPTH improved subchondral bone microarchitecture during osteoarthritis progression

To examine the effect of iPTH on subchondral bone architecture in osteoarthritis, we performed morphometric analysis on subchondral bone using 3-dimensional (3-D) μCT analysis (*Figure 3A*). Trabecular bone pattern factor (Tb.pf), a measure of disruption in trabecular bone connectivity (*Hahn et al., 1992*), increased rapidly by week 4 and plateaued in vehicle mice after DMM surgery (*Figure 3B*), indicating uncoupled subchondral bone remodeling. Tb.Pf in PTH mice after DMM surgery was no different from sham mice within 2 weeks postoperatively, but increased by week 8. However, Tb.Pf of PTH mice was significantly lower than that of vehicle mice (*Figure 3B*), which indicates that early initiation of iPTH slowed the disruption of subchondral trabecular bone connectivity in osteoarthritis. Structure model index (SMI), a measure to quantify the architectural type of cancellous bone (*Hildebrand and Rüegsegger, 1997*), increased significantly by week 4. This increase was sustained in vehicle mice at week 8 after DMM surgery (*Figure 3C*), indicating a change from plate-predominant to rod-predominant cancellous bone architecture in the development of osteoarthritis. SMI in PTH mice also increased after DMM surgery compared with sham surgery, but the magnitude of increase was less than that of vehicle mice at every time point (*Figure 3C*). This difference indicates that early initiation of iPTH slowed the architectural deterioration of subchondral bone in osteoarthritis. Total volume of pore space (Po.V(tot)) and subchondral bone plate thickness (SBP.Th) increased gradually in vehicle mice and PTH mice after DMM surgery compared with sham mice, but the magnitudes of increase in PTH mice were significantly lower than those in vehicle mice at every

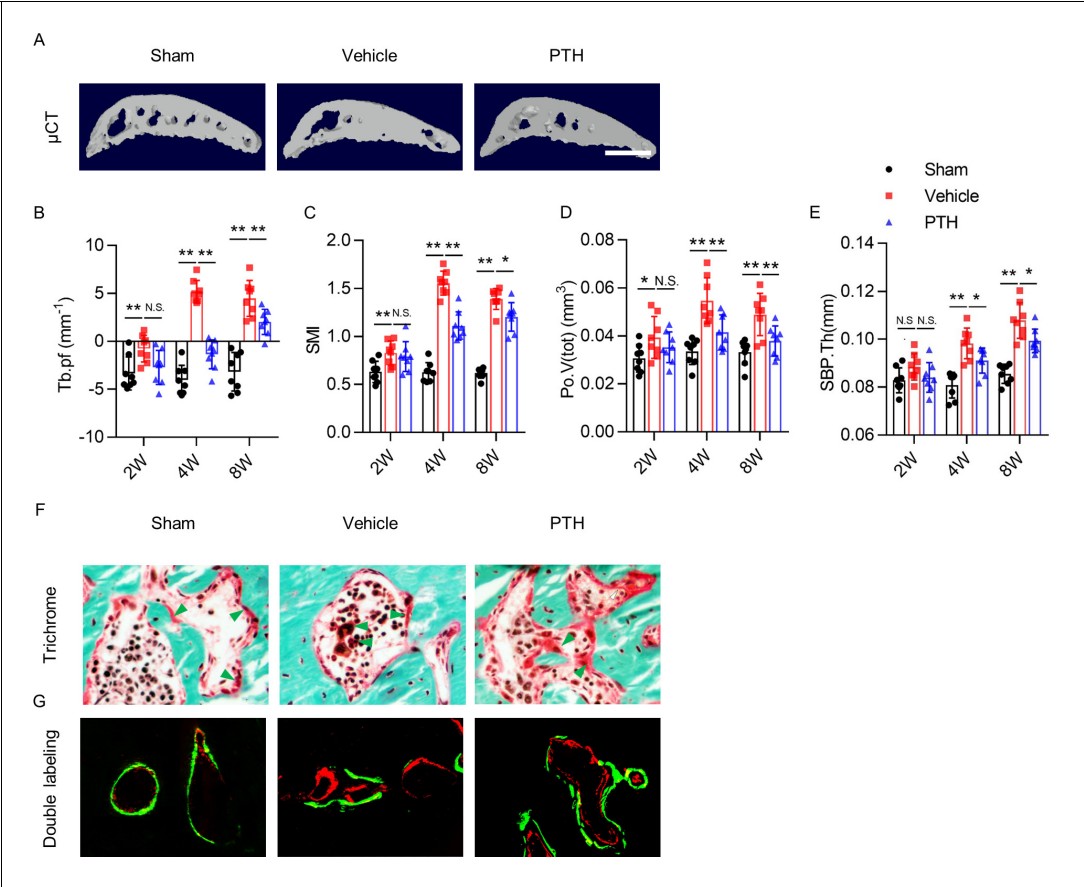

**Figure 3.** PTH sustains subchondral bone microarchitecture by remodeling. (**A**) 3-D, high-resolution microcomputed tomography (μCT) images of the tibial subchondral bone medial compartment (sagittal view) at week 8 after sham or DMM surgery. Scale bar: 500 μm (**B–E**) Quantitative analysis of structural parameters of subchondral bone by μCT analysis: trabecular pattern factor (Tb.pf), structure model index (SMI), total volume of pore space (Po.V(tot)), and thickness of subchondral bone plates (SBP.Th). (n = 8/group). (**F**) Trichrome staining in the tibial subchondral bone sections at week 8 after sham or DMM surgery. Arrowhead indicating the osteoid. Scale bar: 50 μm. (**G**) Calcein (green) and Alizarin (red) fluorescent double labeling of the subchondral bone at week 8 after sham or DMM surgery. Scale bar: 50 μm. *p<0.05, **p<0.01.

The online version of this article includes the following source data for figure 3:

**Source data 1.** Raw data of microCT data.

time point (*Figure 3D,E*). These results indicate that early initiation of iPTH slowed the deterioration and hypertrophy of subchondral bone microarchitecture in osteoarthritis.

To investigate how iPTH improves subchondral bone microarchitecture, we assessed the localization of osteoid formation by performing trichrome staining and Calcein-Alizarin double labeling of tibial subchondral bone sections. Trichrome staining showed that osteoids formed islets in the subchondral bone marrow of DMM mice. iPTH induced formation of osteoids on subchondral bone surface in DMM mice, similar to sham mice (*Figure 3F*). The result was validated in fluorescent double-labeling experiment (*Figure 3G*). These results suggest that iPTH plays an essential role in sustaining coupled subchondral bone remodeling in osteoarthritis.

## iPTH downregulates signaling of elevated active TGF-β by inducing endocytosis of TβRII

To investigate the mechanism by which iPTH maintains coupled subchondral bone remodeling, we analyzed the effects of iPTH on the subchondral organization of osteoprogenitors, osteoclasts, and TGF-β signaling. Immunostaining for Nestin, a marker for adult bone marrow MSCs, showed a significantly higher number of Nestin[+] cells in the subchondral bone marrow of vehicle mice compared with sham mice (*Figure 4A,B*). Once committed to the osteoblast lineage, MSCs express Osterix, a

**Figure 4.** PTH sustains subchondral bone remodeling by endocytosis of TβRII. (**A, B**) Immunofluorescence analysis and quantification of Nestin[+] cells (green) in tibial subchondral bone at week 4 after sham or DMM surgery. Scale bar: 50 μm (n = 8/group). (**C, D**) Immunohistochemical analysis and quantification Osterix[+] cells (brown) in tibial subchondral bone in different groups at week 4 after sham or DMM surgery. Scale bar: 50 μm (n = 8/group). (**E, F**) Immunohistochemical analysis and quantification of pSmad2/3[+] cells (brown) in tibial subchondral bone at week 4 after sham or DMM surgery. Scale bar: 50 μm (n = 8/group). (**G, H**) Tartrate-resistant acid phosphatase (TRAP) staining (pink) and quantitative analysis of TRAP[+] cells in tibial subchondral bone at week 4 after sham or DMM surgery. Scale bar: 100 μm (n = 8/group). (**I**) Quantitative analysis of active TGF-β in serum of mice at week 4 after sham or DMM surgery, determined by ELISA (n = 8/group). (**J**) Immunofluorescent analysis of TβRII (green) distribution on mouse bone marrow MSCs. Actin (red); DAPI stains nuclei blue. Scale bar: 10 μm. (**K**) Immunofluorescent analysis of pSmad2/3[+] on mouse bone marrow MSCs. Scale bar: 25 μm. DAPI stains nuclei blue. *p<0.05, **p<0.01.

The online version of this article includes the following source data for figure 4:

**Source data 1.** Raw data of nestin staining, osterix staining, psmad2/3 staining, TRAP staining, and level of active TGF-β.

marker of osteoprogenitors. The number of Osterix[+] osteoprogenitors was also greater in the subchondral bone marrow of vehicle mice compared with sham mice (*Figure 4C,D*). Intermittent PTH attenuated the increases of Nestin[+] and Osterix[+] cells in subchondral bone marrow of DMM mice (*Figure 4A–D*), suggesting that early initiation of iPTH attenuates aberrant bone formation in osteoarthritis by modulating the recruitment of osteoprogenitors in subchondral bone.

Immunostaining of pSmad2/3 revealed that the number of pSmad2/3$^+$ cells in subchondral bone increased significantly after DMM surgery compared with sham surgery, but this effect was significantly attenuated in PTH mice (*Figure 4E,F*). Tartrate-resistant acid phosphatase$^+$ (TRAP$^+$) staining showed that the number of osteoclasts increased significantly in subchondral bone 4 weeks post-DMM surgery; interestingly, iPTH treatment further increased the number of TRAP$^+$ osteoclasts in subchondral bone (*Figure 4G,H*). Furthermore, ELISA of active TGF-β1 in serum revealed that active TGF-β1 concentration increased after DMM surgery compared with sham surgery, and iPTH further increased the active TGF-β1 concentration in DMM mice compared with vehicle mice (*Figure 4I*). These results indicate that early initiation of iPTH attenuates aberrant subchondral bone remodeling by interfering with downstream TGF-β signaling in osteoarthritis.

PTH1R and TβRII form an endocytic complex in response to PTH (*Qiu et al., 2010*). To explore the mechanism by which PTH interferes with TGF-β downstream signaling, we analyzed the localization of TβRII on MSCs. TβRII was localized mainly at the MSC surface membrane in the negative control group and vehicle group, and the amount of cell-surface TβRII decreased significantly after PTH stimulation (*Figure 4J*). Moreover, with stimulation of TGF-β1, immunostaining of pSmad2/3 showed that phosphorylation and nuclear accumulation of Smad2/3 increased dramatically in the vehicle group compared with the negative control group; however, PTH decreased TGF-β1-induced phosphorylation and the nuclear accumulation of Smad2/3 compared with vehicle treatment in MSCs (*Figure 4K*). Collectively, these results indicate that aberrant subchondral bone formation due to elevated TGF-β1 signaling in osteoarthritis was attenuated in the setting of PTH-induced endocytosis of TβRII.

## iPTH attenuates osteoarthritic pain and progression of joint degeneration

We tested a delayed regimen of starting iPTH to examine the effect of iPTH in a situation that mimics the clinical situation where therapy is initiated after the diagnosis of osteoarthritis. In this delayed iPTH treatment regimen, iPTH was initiated 4 weeks after DMM surgery. Delayed iPTH attenuated the DMM-associated decreases in 50% PWT and PAMWT (*Figure 5A,B*). Gait analysis showed that iPTH attenuated the DMM-associated decreases in affected hind paw intensity, contact area, and swing speed relative to the contralateral side (*Figure 5C*). The immunostaining of tibial subchondral bone sections revealed that iPTH significantly ameliorated the DMM-associated increases in the relative density of CGRP$^+$ sensory nerve fibers in subchondral bone (*Figure 5D,E*). ELISA analysis of DMM mice showed that delayed iPTH decreased the level of PGE2 in subchondral bone compared with vehicle-treated mice (*Figure 5F*). Delayed iPTH attenuated the DMM-associated degeneration of articular cartilage as reflected by SOFG staining and OARSI scores (*Figure 5G, H*). Similar to the early initiation of iPTH, a delayed regimen of iPTH significantly attenuated microarchitectural deterioration of subchondral bone after DMM surgery compared with vehicle mice (*Figure 5I*). µCT analysis showed that delayed iPTH decreased Tb.Pf, SMI, Po.V(tot), and SPB.Th compared with vehicle mice (*Figure 5J*). These results indicate that iPTH can treat osteoarthritis pain and progression in later stages of the disease by attenuating nociceptive nerve innervation in subchondral bone and by preventing articular cartilage degeneration and deterioration of the subchondral bone microstructure.

## Knockout of PTH1R in MSCs blunts the effects of iPTH on osteoarthritis pain and degeneration

To validate the role of PTH-induced remodeling of subchondral bone in the attenuation of osteoarthritis pain and osteoarthritis progression, we induced conditional knockout of PTH1R in Nestin$^+$ MSCs of DMM mice. Nestin-Cre$^{ERT2}$::*Pth1r*$^{fl/fl}$ (*Pth1r*$^{-/-}$) mice were injected with tamoxifen to delete PTH1R in the Nestin$^+$ MSCs. Intermittent PTH had no effect on osteoarthritis pain when PTH1R was conditionally knocked out in MSCs, as reflected by no differences in 50% PWT, PAMWT, or gait analysis between iPTH–treated and vehicle-treated *Pth1r*$^{-/-}$ mice after DMM surgery, whereas the effect of iPTH on reducing osteoarthritis pain was re-demonstrated in *Pth1r*$^{fl/fl}$ (*Pth1r*$^{+/+}$) mice (*Figure 6A–C*). This result indicates that iPTH attenuates osteoarthritis pain by signaling through PTH1R in Nestin$^+$ MSCs in subchondral bone. Moreover, in DMM *Pth1r*$^{-/-}$ mice, iPTH was ineffective in reduction of subchondral sensory innervation, such as CGRP$^+$ nociceptive nerve fibers, in coupling with

**Figure 5.** Delayed PTH attenuates progressive osteoarthritis pain and joint degeneration in DMM model mice. (**A, B**) 50% PWT at the left hind paw (LH) and PAMWT at the left knee in PTH-treated DMM and vehicle-treated DMM mice, starting from week 4 to week 8 after DMM surgery (n = 8/group). *, **Vehicle-treated DMM mice compared with PTH-treated DMM mice. (**C**) Quantitative analysis of LH intensity, LH area, and LH swing speed compared with RH, based on catwalk analysis (n = 8/group). (**D**) Immunofluorescent analysis of the density of CGRP[+] (green) sensory nerve fibers in tibial subchondral bone of sham-operated, PTH-treated DMM, and vehicle-treated mice at week 8 after sham or DMM surgery. DAPI stains nuclei blue. Scale bar: 50 μm. (**E**) Quantitative analysis of the density of CGRP[+] sensory nerve fibers in tibial subchondral bone after DMM surgery (n = 8/group). (**F**) Quantitative analysis of PGE2 in subchondral bone determined by ELISA (n = 8/group). (**G**) SOFG staining of sagittal sections of the tibia medial compartment, proteoglycan (red) and bone (green) at week 8 after sham or DMM surgery. Scale bar: 250 μm (n = 8/group). (**H**) OARSI scores (n = 8/group). (**I**) 3-D, high-resolution μCT images of the tibial subchondral bone medial compartment (sagittal view) at week 8 after sham or DMM surgery. Scale bar: 500 μm. (**J**) Quantitative analysis of structural parameters of subchondral bone by μCT analysis: Tb.pf, SMI, Po.V(tot), and SBP.Th (n = 8/group). *p<0.05, **p<0.01.

The online version of this article includes the following source data for figure 5:

**Source data 1.** Raw data of PAMWT, 50% PWT, catwalk analysis, CGRP staining, quantification of level of PGE2, OARSI, and microCT data.

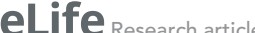

**Figure 6.** PTH-induced osteoarthritis pain relief is inhibited by PTH type one receptor knockout on Nestin[+] MSCs. (A, B) 50% PWT at the LH and PAMWT at the left knee in sham-operated, PTH-treated DMM, and vehicle-treated DMM PTH type one receptor[−/−] (*Pth1r[−/−]*) and *Pth1r[+/+]* mice at week 8 after sham or DMM surgery (n = 8/group). (C) Quantitative analysis of LH intensity, LH area, and LH swing speed compared with the RH in sham-operated, PTH-treated DMM, and vehicle-treated DMM *Pth1r[−/−]* and *Pth1r[+/+]* mice at week 8 after sham or DMM surgery, based on catwalk
*Figure 6 continued on next page*

**Figure 6 continued**

analysis (n = 8/group). (**D–F**) Immunofluorescent and quantitative analysis of CGRP[+] (green) sensory nerve fibers and EMUN+ vessels (red) in tibial subchondral bone of PTH-treated or vehicle-treated *Pth1r*[−/−] and *Pth1r*[+/+] mice at week 8 after DMM surgery (n = 8/group). Scale bar: 50 μm. (**G, H**) Immunofluorescent and quantitative analysis of CGRP[+] (green) sensory nerve fibers in synovium of PTH-treated or vehicle-treated *Pth1r*[−/−] and *Pth1r*[+/+] mice at week 8 after DMM surgery (n = 8/group). Scale bar: 50 μm. (**I**) μCT images of the tibial subchondral bone medial compartment (coronal) at week 8 after DMM surgery. Arrowhead indicates the osteophyte. Scale bar: 500 μm. (**J**) Total volume measurement of osteophytes from the tibial plateau of sham-operated, PTH-treated DMM, and vehicle-treated DMM *Pth1r*[−/−] and *Pth1r*[+/+] mice (n = 8/group). (**K**) Immunohistochemical quantification of COX2[+] cells in the tibial subchondral bone of mice at week 4 after DMM surgery (n = 8/group). (**L**) Quantitative analysis of PGE2 in subchondral bone determined by enzyme-linked immunosorbent assay (ELISA) (n = 8/group). *p<0.05, **p<0.01. NS, no significant difference.

The online version of this article includes the following source data and figure supplement(s) for figure 6:

**Source data 1.** Raw data of PAMWT, 50% PWT, catwalk analysis, CGRP staining, EMUN staining, CGRP staining in synovium, quantification of osteophyte volume, and COX2 staining, and quantification of level of PGE2.

**Figure supplement 1.** Expression levels of SP, P2X3 and PiEZO in nerve fibers were signigficantly decreased PTH1R knockout mice.

**Figure supplement 1—source data 1.** Raw data of SP staining, P2 × 3 staining, and PIEZO2 staining.

decreased EMUN[+] vessels post-DMM surgery (*Figure 6D–F*). Additionally, iPTH was ineffective at reducing subchondral sensory innervation, such as SP[+], P2 × 3[+], and PIEZO2[+] nociceptive nerve fibers after DMM surgery (*Figure 6—figure supplement 1A–E*). Interestingly, the relative densities of CGRP[+] sensory nerve fibers in the joint synovium were unaffected in DMM *Pth1r*[−/−] mice, with or without iPTH treatment after DMM surgery (*Figure 6G,H*). Additionally, there were no obvious difference in osteophyte volume among these four DMM groups (*Figure 6I,J*). Furthermore, iPTH failed to decrease the number of COX2[+] cells and the level of PGE2 in tibial subchondral bone in *Pth1r*[−/−] mice (*Figure 6K,L*), indicating that iPTH modulated the level of PGE2 in subchondral bone in osteoarthritis through its effects on subchondral bone. These results indicate that iPTH attenuates osteoarthritis pain by reducing sensory innervation and PGE2 expression in subchondral bone.

Similarly, iPTH failed to prevent joint degeneration in *Pth1r*[−/−] mice after DMM surgery. As reflected by SOFG staining and OARSI scores, despite less protection of proteoglycan loss of articular cartilage by iPTH in *Pth1r*[−/−] mice compared with *Pth1r*[+/+] mice, iPTH still significantly attenuated the DMM-associated degeneration of articular cartilage in *Pth1r*[−/−] mice compared with vehicle-treated *Pth1r*[−/−] mice (*Figure 7A,B*). In *Pth1r*[−/−] mice, iPTH was ineffective at maintaining the microarchitecture of tibial subchondral bone after DMM surgery, including Tb.Pf, SMI, Po.V(tot), and SBP. Th (*Figure 7C,D*), but the protective effect of iPTH was re-demonstrated in *Pth1r*[+/+] mice after DMM surgery, suggesting that iPTH maintains the subchondral microarchitecture in osteoarthritis via its effect on MSCs. The effect of iPTH on decreasing the number of pSmad2/3[+] cells in tibial subchondral bone after DMM was abolished in *Pth1r*[−/−] mice compared with *Pth1r*[+/+] mice (*Figure 7E, F*). In addition, iPTH failed to decrease the number of Nestin[+] MSCs and Osterix[+] osteoprogenitors in the subchondral bone marrow in *Pth1r*[−/−] mice compared with *Pth1r*[+/+] mice after DMM surgery (*Figure 7G–J*). These findings support the notion that PTH sustains subchondral bone remodeling through inhibition of excessive active TGF-β signaling and suggest that the roles of PTH in the attenuation of osteoarthritis pain and the sustaining of subchondral bone microarchitecture are derived mainly from its role in subchondral bone.

## Discussion

The current treatments for osteoarthritis pain achieve limited therapeutic effects, and they are palliative for progressive pathological joint changes (*Hochberg et al., 2012*). Although analgesics and nonsteroidal anti-inflammatory drugs were recommended in the 2012 American College of Rheumatology guidelines, these treatments produce unsustained and insufficient control of osteoarthritis pain with substantial adverse effects and fail to attenuate osteoarthritis progression effectively (*Hochberg et al., 2012*). Joint replacement is the only alternative for end-stage osteoarthritis (*Thomas et al., 2009*). Therefore, the development of an effective disease-modifying treatment for osteoarthritis is needed. We found that PTH reduced sensory innervation and the level of PGE2 in subchondral bone through inhibition of aberrant subchondral bone remodeling, which resulted in osteoarthritis pain relief. Importantly, PTH also attenuated osteoarthritis progression by inhibiting deterioration of subchondral bone microarchitecture and cartilage degeneration (*Figure 8*).

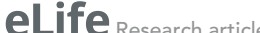

**Figure 7.** PTH-induced bone remodeling is inhibited by PTH1R knockout on Nestin[+] MSCs. (**A**) SOFG staining of sagittal sections of the tibial medial compartment, proteoglycan (red) and bone (green) at week 8 after DMM surgery. Scale bar: 250 µm. (**B**) OARSI score (n = 8/group). (**C**) 3-D, high-resolution µCT images of the tibial subchondral bone medial compartment at week 8 after DMM surgery. Scale bar: 500 µm. (**D**) Quantitative analysis of structural parameters of subchondral bone by µCT analysis: Tb.pf, SMI, Po.V (tot), and SBP.Th (n = 8/group). (**E, F**) Immunohistochemical analysis and

*Figure 7 continued on next page*

Figure 7 continued

quantification of pSmad2/3$^+$ cells in subchondral bone marrow at week 4 after DMM surgery. Scale bar: 50 μm (n = 8/group). (G, H) Immunofluorescent analysis and quantification of Nestin$^+$ cells (green) in tibial subchondral bone marrow at week 4 after DMM surgery. Scale bar: 50 μm; n = 8/group. (I, J) Immunohistochemical analysis and quantification of Osterix$^+$ (brown) cells in tibial subchondral bone marrow at week 4 after DMM surgery. Scale bar: 50 μm; n = 8/group. *p<0.05, **p<0.01. NS, no significant difference.

The online version of this article includes the following source data for figure 7:

**Source data 1.** Raw data of OARSI, microCT, psmad2/3 staining, nestin staining, and osterix staining.

Osteoarthritis pain originates primarily from synovium, ligaments, menisci, subchondral bone, and muscle and joint capsules (*Mathiessen and Conaghan, 2017*; *Reimann and Christensen, 1977*; *Ashraf et al., 2011*; *Kc et al., 2016*; *Hirasawa et al., 2000*; *Belluzzi et al., 2019*). In general, PTH attenuate osteoarthritis by maintaining microarchitecture of the subchondral bone and protection of articular cartilage (*Sampson et al., 2011*; *Orth et al., 2013*; *Bellido et al., 2011*; *Yan et al., 2014*; *Chang et al., 2009*; *Lugo et al., 2012*; *Morita et al., 2018*; *Dai et al., 2016*; *Chen et al., 2018*; *Petersson et al., 2006*). We have previously shown that aberrant subchondral bone is a critical source of pain in osteoarthritis (*Zhu et al., 2019*). Abnormal subchondral bone remodeling induces aberrant sensory innervation (*Zhu et al., 2019*), and elevated PGE2 induces hypersensitivity through sensory nerves (*Ni et al., 2019*). In this study, PTH improved primary and secondary knee hyperalgesia and gait deficits in DMM mice. Furthermore, PTH effectively reduced the density of sensory nerve in the subchondral bone in DMM mice. In the *Pth1r$^{-/-}$* mice after DMM surgery, PTH failed to improve osteoarthritis pain and reduce the sensory innervation and the level of PGE2 in the subchondral bone, but PTH worked effectively in *Pth1r$^{+/+}$* mice. We found that the density of CGRP$^+$ sensory nerve fibers in the synovium increased post DMM surgery. This could be the reason that the pain behavior was not completely disappeared in the PTH treatment mice. The density of CGRP$^+$ sensory nerve fibers in the synovium remained unchanged after PTH injection. PTH has been shown to inhibit the expression of pro-inflammatory modulators, including COX2, in the synovial membrane of the osteoarthritis animal models (*Lugo et al., 2012*). Also, the inflammatory response of synovial membrane was also reported to remain unaffected with PTH treatment (*Orth et al., 2013*). The pain that originates from synovium partially reflects the severity of osteoarthritis and further limits overuse of knee joint. Considering that the articular cartilage has no nerve and blood vessel (*Schaible, 2012*), PTH-induced decrease of osteoarthritis pain is primarily due to its effect on subchondral bone.

Blood vessels and nerve fibers often course alongside each other because they share similar mechanisms of wiring (*Carmeliet and Tessier-Lavigne, 2005*); therefore, nerve fibers may undergo similar remodeling pattern to vessels with iPTH treatment. Co-immunostaining showed that CGRP$^+$ nerve fibers is largely colocalized with endomucin+ vessels in the subchondral bone, both significantly decreased with iPTH treatment. The expression of PTH1R in arteriole (*Massfelder et al., 2002*; *Song et al., 2009*) and endothelial cells (*Funk et al., 2002*; *Isales et al., 2000*) has been reported in previous studies. Noteworthy, endogenous PTHrP downregulates the expression of PTH1R in vascular smooth muscle cells (*Song et al., 2009*). As we all know, the vessel volume is significantly increased in the subchondral bone in osteoarthritis (*Zhen et al., 2013*). PTH may inhibit vascularization or angiogenesis by negatively modulating the expression of PTH1R in the subchondral bone, where the vessel formation is usually accompanied by nerve growth. Thus, PTH reduces sensory innervation in the subchondral bone in association with decrease of blood vessels.

Sensory innervation promotes osteoblast bone formation and suppresses osteoclast bone resorption (*Li et al., 2017*; *Ishizuka et al., 2005*). We have shown that sensory nerve regulates bone homeostasis and promote regeneration (*Chen et al., 2019*). The ablation of sensory innervation by genetic or pharmacological approaches consistently results in decreased bone mass in adult mice (*Chen et al., 2019*; *Fukuda et al., 2013*; *Brazill et al., 2019*). PTH induces osteoclastic bone resorption in coupling osteoblast bone formation during bone remodeling. Increased osteoclasts secrete PDGF-BB for type H vessel formation in support of the bone remodeling (*Xie et al., 2014*). Accumulatively, the daily injection of PTH increases bone formation with net decrease of both total vessel formation and nerve innervation as the porous subchondral bone was decreased with new bone formation during the dynamic process. Therefore, iPTH decreased sensory nerve and attenuated osteoarthritis pain by a decrease of porous subchondral bone.



**Figure 8.** Schematic diagram of PTH-induced pain relief and attenuation of joint degeneration in osteoarthritis. Aberrant mechanical stress induces uncoupled remodeling of subchondral bone due to excessive TGF-β at onset of osteoarthritis and subsequently generates a pathological microenvironment with significantly increased PGE2 level and other inflammatory factors when reaching a certain threshold. Additionally, aberrant microarchitecture is associated with increased wiring of sensory nerve fibers and vessels in the subchondral bone. PTH reduced sensory innervation, vessel wiring, and the level of PGE2 by maintaining the microarchitecture of subchondral bone through induction of endocytosis of PTH1R and TβRII.

The abnormal microarchitecture of subchondral bone caused by aberrant subchondral bone remodeling plays a causal role in the pathogenesis of osteoarthritis (*Zhen et al., 2013*; *Pan et al., 2012*). The maintenance of active TGF-β levels in a spatial and temporally manner is essential for coupled bone remodeling (*Tang et al., 2009*), but excessive active TGF-β signaling in the subchondral bone leads to aberrant bone remodeling (*Zhen et al., 2013*). PTH has been shown to preserve microarchitecture of subchondral bone and improves subchondral bone reconstitution of osteochondral defects (*Orth et al., 2013*; *Bellido et al., 2011*; *Yan et al., 2014*; *Morita et al., 2018*;

*Dai et al., 2016*). The decreased bone density and aberrantly elevated TGF-β levels were simultaneously occurred in the subchondral bone at the early stage of osteoarthritis (*Zhen et al., 2013*). PTH induces internalization of PTH1R together with TβRII (*Qiu et al., 2010*). iPTH improves the pathological subchondral bone microenvironment by downregulating the TGF-β signaling through endocytic complex of PTH1R and TβRII.

Aberrant mechanical stress induces subchondral bone uncoupled remodeling at the onset of osteoarthritis and subsequently generates a pathological microenvironment with significantly increased PGE2 level and other inflammatory factors when reaching a certain threshold (*Rahmati et al., 2016*). Importantly, subchondral bone has been shown as a source of inflammatory mediators and osteoarthritis pain. The abnormal microarchitecture of subchondral bone in osteoarthritis dramatically changes the stress distribution on articular cartilage. iPTH modulates the mechanical stress distribution on articular cartilage by improving microarchitecture of the subchondral bone. As a result, the production of pro-inflammatory factors, such as PGE2, in subchondral bone is reduced. iPTH also has been shown to ameliorate both hyperplasia and fibrosis in osteoarthritis preceded by osteoporosis and inhibit expression of pro-inflammatory modulators, including COX2, in synovial membrane (*Lugo et al., 2012*). We revealed that iPTH have a therapeutic effect on osteoarthritis and pain by improving subchondral bone microenvironment through remodeling. For example, unmineralized or low mineralized bony tissues in the subchondral bone marrow cavity, not on the bone surface, were the typical osteoarthritis pathological change (*Zhen et al., 2013*). The newly formed osteoid islets in the subchondral bone marrow were almost diminished in osteoarthritis mice with daily PTH injection.

Articular cartilage degeneration is the primary concern in osteoarthritis. PTH has been reported to inhibit cartilage degradation, terminal differentiation, and apoptosis of chondrocytes and to promote regeneration of articular cartilage in osteoarthritis (*Sampson et al., 2011*; *Orth et al., 2013*; *Bellido et al., 2011*; *Yan et al., 2014*; *Chang et al., 2009*; *Lugo et al., 2012*; *Chen et al., 2018*; *Petersson et al., 2006*). The studies on PTH effect on osteoarthritic subchondral bone showed iPTH administration resulted in improvement of subchondral bone structure (*Bellido et al., 2011*; *Yan et al., 2014*; *Dai et al., 2016*; *Orth et al., 2013*) or contradict result of induction of osteoarthritis (*Orth et al., 2014*). PTH effect on the subchondral bone remains unclear. The current study provides evidence for the mechanism to clarify the different results. The result of PTH induction of osteoarthritis was conducted in normal mice rather than osteoarthritis animal models. In normal mice with no joint osteoarthritis, their subchondral bones have normal bone density and appropriate microstructure, which are in balance with articular cartilage. iPTH treatment generates additional new bone on top of normal subchondral bone density and structure. As a result, the changes of subchondral bone disrupt its balanced interplay with articular cartilage and leads to cartilage degeneration. Whereas in osteoarthritis mice, the subchondral bone is pathologically changed from coupled remodeling to uncoupled remodeling with porous structure. iPTH administration stimulates osteoclast remodeling and generates new bone to improve its structure quality and its interaction with articular cartilage. More importantly, the beneficial effect of PTH on subchondral bone deterioration and pain relief was diminished in the osteoarthritis *Pth1r*⁻/⁻ mice. The protective effects of PTH on cartilage degeneration were also impaired in *Pth1r*⁻/⁻ mice. Thus, our data reveal that PTH protects articular cartilage from degeneration through both articular cartilage and subchondral bone.

## Materials and methods

### Key resources table

| Reagent type (species) or resource | Designation | Source or reference | Identifiers | Additional information |
|---|---|---|---|---|
| Antibody | Rabbit polyclonal to pSmad2/3 | Santa Cruz | sc-11769 | 1:50 |
| Antibody | Rabbit polyclonal to Osterix | Abcam | ab22552 | 1:600 |
| Antibody | Rabbit polyclonal to TβRII | Abcam | ab186838 | 1:100 |
| Antibody | Mouse monoclonal to β-actin | Cell Signaling Technology | CST3700 | 1:3000 |

*Continued on next page*

*Continued*

| Reagent type (species) or resource | Designation | Source or reference | Identifiers | Additional information |
| --- | --- | --- | --- | --- |
| Antibody | Rabbit polyclonal to COX2 | Abcam | ab15191 | 1:100 |
| Antibody | Rabbit polyclonal to MMP13 | Abcam | ab39012 | 1:200 |
| Antibody | Chicken polyclonal to Nestin | Aves Labs | NES0407 | 1:300 |
| Antibody | Mouse monoclonal to CGRP | Abcam | ab81887 | 1:200 |
| Antibody | Rat monoclonal to Substance P | Santa Cruz | sc-21715 | 1:200 |
| Antibody | Rat monoclonal to endomucin | Santa Cruz | sc-65495 | 1:50 |
| Antibody | Rabbit polyclonal to PIEZO2 | Abcam | ab243416 | 1:300 |
| Antibody | Rabbit polyclonal to P2 × 3 | Abcam | ab10269 | 1:500 |
| Strain, strain background (*Mus musculus*) | C57BL/6J mice | Charles River Laboratories | Strain Code: 27 | C57BL/6 background |
| Strain, strain background (*Mus musculus*) | Pth1r$^{fl/fl}$ | *Kobayashi et al., 2002* | Charles River Laboratories | C57BL/6 background |
| Strain, strain background (*Mus musculus*) | Nestin-creERT2 | Stock No: 016261 | Jackson Laboratory | C57BL/6 background |
| Chemical compound, drug | human PTH (1-34) | Sigma-Aldrich | P3796 | N/A |
| Sequence- based reagent | Nestin-creERT2 forward | N/A | PCR Primer | 5′–3′: ACCAGAGACGG AAATCCATCGCTC |
| Sequence-based reagent | Nestin-creERT2 reverse | N/A | PCR Primer | 5′–3′: TGCCACGACCA AGTGACAGCAATG |
| Sequence-based reagent | Pth1r loxP allele forward | N/A | PCR Primer | 5′–3′: TGGACGCAGAC GATGTCTTTACCA |
| Sequence-based reagent | Pth1r loxP allele reverse | N/A | PCR Primer | 5′–3′: ACATGGCCATGC CTGGGTCTGAGA |
| Software, algorithm | Graphpad 8.0 | N/A | Statistical Analysis | Graph preparation |
| Software, algorithm | SPSS, 15.0 | N/A | Statistical Analysis | Statistical analysis |

## Mice and in vivo treatment

We purchased male C57BL/6J mice (wild-type [WT] mice) from Charles River Laboratories (Wilmington, MA). We anesthetized mice at 10 weeks of age with xylazine (Rompun, Sedazine, AnaSed; 10 mg/kg, intraperitoneally) and ketamine (Vetalar, Ketaset, Ketalar; 100 mg/kg, intraperitoneally). Then, the DMM model was created by transecting the meniscotibial ligament that connects the lateral side of the medial meniscus with the intercondylar eminence of the tibia to induce instability of the left knee. Sham DMM operations were performed by opening the joint capsule of the left knee of independent mice. Sham mice or DMM mice were injected subcutaneously with 40 μg/kg per day of human PTH (*Centers for Disease Control and Prevention (CDC), 2009*; *Murray et al., 2012*; *Peat et al., 2001*; *Lane et al., 2010*; *Geba et al., 2002*; *Karlsson et al., 2002*; *Mathiessen and Conaghan, 2017*; *Malfait and Schnitzer, 2013*; *Dieppe and Lohmander, 2005*; *Hannan et al., 2000*; *Bedson and Croft, 2008*; *Yusuf et al., 2011*; *Kwoh, 2013*; *Laslett et al., 2012*; *Laslett et al., 2014*; *Isaac et al., 2005*; *Reilly et al., 2005*; *Bhosale and Richardson, 2008*; *Zhen and Cao, 2014*; *Zhen et al., 2013*; *Zhu et al., 2019*; *Ni et al., 2019*; *Tang et al., 2009*; *Crane and Cao, 2014*; *Qiu et al., 2010*; *Okabe and Graham, 2004*; *Sampson et al., 2011*; *Orth et al., 2013*; *Bellido et al., 2011*; *Yan et al., 2014*; *Chang et al., 2009*; *Lugo et al., 2012*; *Eswaramoorthy et al., 2012*; *Cui et al., 2020*) (Sigma–Aldrich, St. Louis, MO) or the equivalent volume of vehicle (phosphate-buffered saline [PBS]). For immediate regime of administration, daily injection of PTH was initiated 3 days after DMM surgery. The operated mice were euthanized at 2, 4, or 8 weeks after surgery (n = 8–12 mice per group). For delayed regime of administration, daily injection of PTH was initiated 4 weeks after DMM surgery (n = 8–12 mice per group).

We purchased Nestin-cre$^{ERT2}$ mice from The Jackson Laboratory (Bar Harbor, ME). Mice with floxed *Pth1r* (*Pth1r$^{fl/fl}$*) were obtained from the laboratory of Dr. Henry Kronenberg (*Kobayashi et al., 2002*). Heterozygous male Nestin-cre$^{ERT2}$ mice were crossed with *Pth1r$^{fl/fl}$* mice. The offspring were intercrossed to generate the following genotypes: WT mice, Nestin-cre$^{ERT2}$ mice,

$Pth1r^{fl/fl}$(herein, $Pth1r^{+/+}$) mice, and Nestin-cre$^{ERT2}$::$Pth1r^{fl/fl}$ mice (herein, $Pth1r^{-/-}$), in which *Cre* was fused with a mutated estrogen receptor that could be activated by tamoxifen. We determined the genotype of transgenic mice by polymerase chain reaction (PCR) analyses of genomic DNA isolated from mouse tails. The required primers were as follows: *Pth1r*: Forward 5′−3′: TGGACGCAGACGA TGTCTTTACCA, Reverse 5′−3′: ACATGGCCATGCCTGGGTCTGAGA; Cre: Forward 5′−3′: ACCA-GAGACGGAAATCCATCGCTC, Reverse 5′−3′: TGCCACGACCAAGTGACAGCAATG. We performed DMM surgery on 10 week old male $Pth1r^{+/+}$ mice and $Pth1r^{-/-}$ mice. Three days after surgery, we treated each group with tamoxifen (100 mg/kg) daily, and the mice were injected subcutaneously with either PTH (40 µg/kg/day) or the equivalent volume of vehicle (PBS) subcutaneously daily for 4 or 8 weeks (n = 8 mice per treatment group).

Mice were euthanized with an overdose of inhaled isoflurane. We obtained whole blood samples by cardiac puncture immediately after euthanasia. Serum was collected by centrifuge at 200 × g for 15 min and stored at −80°C before analysis. The mice were then flushed with PBS for 5 min, followed by 10% buffered formalin perfusion for 5 min via the left ventricle. Then, the left knees were dissected and fixed in 10% buffered formalin for 48 hr.

## Cell culture

We isolated bone marrow MSCs from male WT mice at 6 weeks of age, as described by *Soleimani and Nadri, 2009*. We maintained cells (passage 3–5) in Iscove's modified Dulbecco's medium (Invitrogen, Carlsbad, CA) supplemented with 10% fetal calf serum (Atlanta Biologicals, Flowery Branch, GA), 10% horse serum (Thermo Fisher Scientific, Waltham, MA), and 1% penicillin–streptomycin (Mediatech, Inc, Manassas, VA). We cultured bone marrow MSCs in six-well plates at a density of 1.8 × $10^5$ cells per well; MSCs were then starved for 12 hr, followed by human PTH (*Centers for Disease Control and Prevention (CDC), 2009*; *Murray et al., 2012*; *Peat et al., 2001*; *Lane et al., 2010*; *Geba et al., 2002*; *Karlsson et al., 2002*; *Mathiessen and Conaghan, 2017*; *Malfait and Schnitzer, 2013*; *Dieppe and Lohmander, 2005*; *Hannan et al., 2000*; *Bedson and Croft, 2008*; *Yusuf et al., 2011*; *Kwoh, 2013*; *Laslett et al., 2012*; *Laslett et al., 2014*; *Isaac et al., 2005*; *Reilly et al., 2005*; *Bhosale and Richardson, 2008*; *Zhen and Cao, 2014*; *Zhen et al., 2013*; *Zhu et al., 2019*; *Ni et al., 2019*; *Tang et al., 2009*; *Crane and Cao, 2014*; *Qiu et al., 2010*; *Okabe and Graham, 2004*; *Sampson et al., 2011*; *Orth et al., 2013*; *Bellido et al., 2011*; *Yan et al., 2014*; *Chang et al., 2009*; *Lugo et al., 2012*; *Eswaramoorthy et al., 2012*; *Cui et al., 2020*) (Sigma–Aldrich, St. Louis, MO) and/or TGF-β1 (R and D Systems, Minneapolis, MN) treatment as indicated.

## Histochemistry, immunohistochemistry, and histomorphometry

The joint samples were decalcified in 10% ethylenediamine tetraacetic acid (EDTA, pH 7.4) for 14 days and embedded in paraffin or Optimal Cutting Temperature Compound (Sakura Finetek, Torrance, CA). Four micrometer thick sagittal-oriented sections of the medial compartment of the knee were processed for Safranin O (Sigma–Aldrich, S2255) and fast green (Sigma–Aldrich, F7252) staining, TRAP staining (Sigma–Aldrich, 387A-1KT), and immunohistochemistry staining using a standard protocol. Ten micrometer thick sagittal-oriented sections were used for Nestin immunofluorescent staining, and 30 µm thick sagittal-oriented sections were used for nerve-related immunofluorescent staining using a standard protocol. The tissue sections were incubated with primary antibodies to mouse pSmad2/3 (1:50, sc-11769, Santa Cruz), Osterix (1:600, ab22552, Abcam), TβRII (1:100, ab186838, Abcam), β-Actin (1:3000, CST3700, Cell Signaling Technology), COX2 (1:100, ab15191, Abcam), MMP13 (1:200, ab39012, Abcam), COLX (1:200, ab58632, Abcam), Nestin (1:300, NES0407, Aves Labs, Inc, Davis, CA), CGRP (1:200, ab81887, Abcam), Substance P(1:200, sc-21715, Santa Cruz), endomucin (1:50, sc-65495, Santa Cruz), PIEZO2 (1:300, ab243416, Abcam), and P2 × 3 (1:500, ab10269, Abcam) overnight at 4°C in a humidifier chamber. For immunohistochemical staining, a horseradish peroxidase–streptavidin detection system (Dako, Agilent Technologies, Santa Clara, CA) was used to detect immunoactivity, followed by counterstaining with hematoxylin (Sigma–Aldrich). For immunofluorescence staining, slides were incubated with corresponding secondary antibody at room temperature for 1 hr while avoiding light. Then, the sections were counterstained with 4′,6-diamidino-2-phenylindole (DAPI, Vector, H-1200, Vector Laboratories, Inc, Burlingame, CA). The sample image were captured with Zeiss LSM 780 confocal microscope (Carl

Zeiss Microscopy, LLC, White Plains, NY) or an Olympus BX51 microscope (Olympus Scientific Solutions Americas Inc, Waltham, MA). Quantitative histomorphometric analysis was performed using Image J software (Media Cybernetics Inc, Rockville, MD) in a blinded fashion. We calculated OARSI scores as previously described (*Pritzker et al., 2006*). For quantification of relative intensity of nerve fiber, positive fluorescence signals of the entire subchondral bone was threshold (threshold range: 50–255) and calculated using Image J (Version 1.49). One section at the similar sagittal location of each mouse (three slices per mouse and eight mice per group) was calculated and normalized to that of sham mice (set average to 1).

A double-labeling procedure was performed to label mineralization deposition. Briefly, we injected the mice subcutaneously with 0.1% Calcein (10 mg/kg, Sigma–Aldrich) and 2% Alizarin red (20 mg/kg, Sigma–Aldrich) in a 2% sodium bicarbonate solution 10 days and 3 days, respectively, before sacrifice. We used a fluorescence microscope to capture labeling images of undecalcified bone slices.

## Behavioral testing

Behavioral tests were performed 2 days before surgery and weekly after surgery. All behavioral tests were performed by the same investigators, who were blinded to the allocation of groups.

Automated gait analysis was performed on walking mice using a 'catwalk' system (CatWalk XT, Noldus Information Technology, Leesburg, VA). All experiments were performed and analyzed as previously reported (*Hamers et al., 2006*; *Hamers et al., 2001*). Gait analysis was performed in a room that was dark except for the light from the computer screen. Briefly, mice were trained to cross the catwalk walkway daily for 7 days before surgery. During the test, each mouse was placed individually in the catwalk walkway and allowed to walk freely from one side to the other side of the walkway's glass plate. Light from an encased fluorescent lamp was emitted inside the glass plate and completely internally reflected. When the mouse paws contacted the glass plate, light was reflected down, and the illuminated contact area was recorded with a high-speed color video camera positioned under the glass plate and connected to a computer running CatWalk XT software, v9.1 (Noldus Information Technology). Comparison was made between the left hind paw and right hind paw in each run of each animal. We calculated the following gait parameters: left hind paw (LH)/right hind paw (RH) contact area, LH/RH intensity, and LH/RH swing speed.

The 50% PWT was measured by von Frey hair algesiometry. Mice were habituated to elevated plexiglass chambers and wire mesh flooring before assessments of allodynia. Then, ipsilateral hind paw mechanosensitivity was measured using a modification of the Dixon up-down method (*Dixon, 1980*). Allodynia was evaluated by applying von Frey hairs in ascending order of bending force (force range: 0.07, 0.40, 0.60, 1.0, 1.4, 2.0, 4.0, or 6.0 g). The von Frey hair was applied perpendicular to the plantar surface of the hind paw (avoiding the toe pads) for 2–3 s. If no response, the next higher strength of hair was used, up to the maximum level of 6 g of bending force. If a withdrawal response occurred, the paw was re-tested, starting with the next descending von Frey hair until no response occurred. Four more measurements were made after the first difference was observed. The 50% PWT was determined by using the following formula:

$$50\%\,\mathrm{PWT} = 10^{\mathrm{Xf}+\mathrm{k}\delta}/10{,}000,$$

where Xf is the exact value (in log units) of the final test of von Frey hair, K is the tabular value for the pattern of the last six positive/negative responses, and $\delta$ is the mean difference (in log units) between stimuli. The threshold force required to elicit paw withdrawal (median 50% withdrawal) was determined twice on each hind paw (and averaged) on each testing day, with sequential measurements separated by at least 10 min.

The withdrawal thresholds in response to direct pressure on the medial part of the left knee were measured as pressure hyperalgesia (*Kim et al., 2011*). A 5 mm diameter sensor tip of an applied force gauge (SMALGO algometer; Bioseb, Pinellas Park, FL) was used for the vocalization thresholds. The pressure force on the knee was increased at 50 g/s until an audible vocalization was made while the mice were restrained gently by the investigator. The curve of the pressure force was recorded by using BIO-CIS software (Bioseb) to ensure the force increased gradually. A cutoff force of 500 g was used to prevent trauma to the joint. The test was performed twice at an interval of 15 min, and the mean value was recorded as the nociceptive threshold.

## μCT

We dissected the left knee of mice free of soft tissue and fixed the samples in 10% buffered formalin for 48 hr. The joint samples were then transferred into PBS and analyzed by high-resolution μCT (SkyScan 1172, Bruker-microCT, Kontich, Belgium). The scanner was set at a voltage of 65 kV, a current of 153 μA, and a resolution of 9 μm per pixel to measure the subchondral bone and joint. Images were reconstructed and analyzed using NRecon v1.6 and CTAn v1.9 (Skyscan US, San Jose, CA), respectively. Sagittal images of the tibial subchondral bone were used to perform 3-D histomorphometric analyses. The region of interest was defined to cover the whole medial compartment of subchondral bone. We used six consecutive images from the medial tibial plateau for 3-D reconstruction and analysis using 3-D model visualization software (CTVol v2.0, Skyscan US). 3-D structural parameters were analyzed: Tb.Pf, SMI, Po.V(tot), and SBP.Th.

## ELISA

The concentrations of active TGF-β1 in serum were determined using the TGF-β1 ELISA Development Kit (MB100B, R and D Systems), according to the manufacturer's instructions.

The level of PGE2 in subchondral bone was determined using the PGE2 Parameter Assay Kit (KGE004B, R and D Systems). For sample preparation, we snap-froze and crushed the subchondral bone in liquid nitrogen, right after tissue harvest and repaid removal of surrounding tissue and cartilage. Then homogenization buffer (0.1 M phosphate, pH 7.4, containing 1 mM EDTA and 10 μM indomethacin) were added (ratio: 5 ml buffer to 1 g tissue). The sample were homogenized with a Polytron-type homogenizer. The tissue homogenates were spun at 8000 × g for 10 min, and the supernatant was collected for ELISA assay, according to the manufacturer's instructions.

## Statistics

All analyses were performed using SPSS, v15.0, software (IBM Corp., Armonk, NY). Data are presented as means ± standard deviations. The data were normally distributed, unless otherwise noted. The PAMWTs and 50% PWTs were analyzed by two-way repeated-measures ANOVA with Bonferroni's post hoc test in immediate and delayed regime of PTH administration.

All other sets of data at 2 weeks, 4 weeks, and 8 weeks were analyzed by two-way ANOVA with Bonferroni's post hoc test. For other comparisons among multiple groups, we used one-way ANOVA with Bonferroni's post hoc test. For all experiments, the level of significance was set at $p < 0.05$. All inclusion and exclusion criteria were pre-established. No statistical method was used to predetermine the sample size. Experiments were randomized, and the investigators were blinded to allocation during experiments and outcomes assessment.

## Study approval

All animal experiments were performed in accordance with NIH policies on the use of laboratory animals. All experimental protocols were approved by the Animal Care and Use Committee of The Johns Hopkins University.

## Acknowledgements

This research was supported by National Institute on Aging of the National Institutes of Health under Award Number R01 AG068997 and P01 AG66603 (to XC). We thank editors Kerry Kennedy, Jenni Weems, and Rachel Box in the editorial office at the Department of Orthopaedic Surgery, The Johns Hopkins University, for editing the manuscript.

## Additional information

### Funding

| Funder | Grant reference number | Author |
|---|---|---|
| National Institutes of Health | P01AG066603 | Xu Cao |
| National Institutes of Health | 1R01 AG068997 | Xu Cao |

The funders had no role in study design, data collection and interpretation, or the decision to submit the work for publication.

## Author contributions
Qi Sun, Conceptualization, Data curation, Formal analysis, Investigation, Methodology, Writing - original draft, Writing - review and editing; Gehua Zhen, Data curation, Investigation, Methodology, Writing - review and editing; Tuo Peter Li, Data curation, Writing - original draft, Writing - review and editing; Qiaoyue Guo, Data curation, Validation, Investigation, Writing - original draft; Yusheng Li, Resources, Software, Formal analysis, Validation; Weiping Su, Peng Xue, Formal analysis, Validation, Methodology; Xiao Wang, Conceptualization, Supervision, Writing - review and editing; Mei Wan, Data curation, Formal analysis, Visualization, Writing - review and editing; Yun Guan, Supervision, Methodology; Xinzhong Dong, Supervision, Methodology, Writing - review and editing; Shaohua Li, Formal analysis, Supervision, Investigation, Writing - review and editing; Ming Cai, Data curation, Formal analysis, Supervision, Writing - review and editing; Xu Cao, Conceptualization, Funding acquisition, Investigation, Writing - original draft, Project administration, Writing - review and editing

## Author ORCIDs
Gehua Zhen (iD) http://orcid.org/0000-0001-7652-6226
Tuo Peter Li (iD) http://orcid.org/0000-0002-4302-9538
Xiao Wang (iD) http://orcid.org/0000-0001-6395-706X
Mei Wan (iD) http://orcid.org/0000-0001-9404-540X
Xinzhong Dong (iD) https://orcid.org/0000-0002-9750-7718
Xu Cao (iD) https://orcid.org/0000-0001-8614-6059

## Ethics
Animal experimentation: Ethics Statement: All animal experiments were approved by the Institutional Animal Care and Use of Johns Hopkins University, School of Medicine. (Protocol number: Mo18M308).

## Decision letter and Author response
Decision letter https://doi.org/10.7554/eLife.66532.sa1
Author response https://doi.org/10.7554/eLife.66532.sa2

## Additional files
### Supplementary files
• Transparent reporting form

### Data availability
All data generated or analysed during this study are included in the manuscript and supporting files.

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
