## [Decision Letter]

**Acceptance summary:**

In this manuscript, you and your group described a project of administrating intermittent parathyroid hormone (iPTH) to treat osteoarthritis (OA) pain by inhibiting subchondral bone deterioration and subchondral sensory innervation in an OA mouse model of destabilized medial meniscus (DMM). You demonstrated that iPTH reverses defects of subchondral bone structure, sensory innervation and increased prostaglandin E2 (PGE2) levels in subchondral bone area in DMM mice. You also found that increased transforming growth factor β (TGF-β) signaling during OA development caused uncoupled subchondral bone remodeling, which is regulated by PTH-induced endocytosis of the type 1 PTH receptor (PTH1R) / type 2 TGF-β type 2 receptor (TGFβR2) complex. iPTH improved subchondral bone microarchitecture, and decreased level of PGE2 and sensory innervation in subchondral bone area of DMM mice. These findings improve our understanding on OA mechanism and treatment.

**Decision letter after peer review:**

Thank you for submitting your work entitled "Parathyroid hormone attenuates osteoarthritis pain by remodeling subchondral bone in mice" for consideration by *eLife*. Your article has been evaluated by Mone Zaidi as the Senior Editor, and a Reviewing Editor.

In this manuscript, the authors described a protocol of administrating intermittent parathyroid hormone (iPTH) to treat osteoarthritis (OA) pain by inhibiting subchondral bone deterioration and subchondral sensory innervation in an OA mouse model of destabilized medial meniscus (DMM). The authors demonstrated that iPTH reverses defects of subchondral bone structure, sensory innervation and increased prostaglandin E2 (PGE2) levels in subchondral bone area in DMM mice. The authors also demonstrated that increased transforming growth factor β (TGF-β) signaling during OA development caused uncoupled subchondral bone remodeling, which is regulated by PTH-induced endocytosis of the type 1 PTH receptor (PTH1R) / type 2 TGF-β type 2 receptor (TGFβR2) complex. iPTH improved subchondral bone microarchitecture, and decreased level of PGE2 and sensory innervation in subchondral bone area of DMM mice.

This manuscript was submitted to another journal and reviewed by their reviewers. The major concerns raised by the other reviewers are the innovation and significance and statistical methods used. In response to the other reviews, the authors have extensively addressed the questions and comments raised.

We have carefully reviewed this manuscript and found that this manuscript could be adequately published in *eLife* after the authors address some additional questions and comments listed below.

1) Please consolidate the following two sentences.

"This difference indicates that early initiation of iPTH slowed the progression of subchondral porosity and hypertrophy in osteoarthritis. These results indicate that early initiation of iPTH slowed the deterioration of subchondral bone microarchitecture in osteoarthritis."

2) Please address the necessity and significance about measuring the osteoid formation and bone formation rates in subchondral bone area after PTH administration in this project.

3) Subsection “iPTH down-regulates signaling of elevated active TGF-β by inducing endocytosis of TβRII”: The timing is importance for changes in osteoclast numbers in subchondral bone area during OA development. The authors need to disclose this information when they describe changes in osteoclast numbers in the revised manuscript.

4) Subsection “iPTH attenuates osteoarthritic pain and progression of joint degeneration”: To mimic the clinical situation, the author initiated iPTH 4 weeks after DMM surgery. The design of this study is importance since it is more relevant to the clinical situation.

5) In this project, the effect of iPTH was eliminated when PTH1R was deleted in Nestin-expressing cells. In addition to subchondral bone, the authors also showed the effect of iPTH in synovial tissue. The authors need to discuss external effect of iPTH besides Nestin-expressing cells.

---

## [Author Response]

We have carefully reviewed this manuscript and found that this manuscript could be adequately published in eLife after the authors address some additional questions and comments listed below.1) Please consolidate the following two sentences."This difference indicates that early initiation of iPTH slowed the progression of subchondral porosity and hypertrophy in osteoarthritis. These results indicate that early initiation of iPTH slowed the deterioration of subchondral bone microarchitecture in osteoarthritis."

As suggested, we have consolidated the following two sentences as “These results indicate that early initiation of iPTH improved the subchondral bone microarchitecture decreased its porosity and hypertrophy in osteoarthritis.”

2) Please address the necessity and significance about measuring the osteoid formation and bone formation rates in subchondral bone area after PTH administration in this project.

Thanks for the Comments. “Measurement of Osteoid formation and Bone formation rates” is not prepared when we organized our manuscript. Both are required by reviewer 1 when the manuscript was submitted to another journal. We did not see obvious necessity and significance of Data. In the revised manuscript, we removed the related data and description.

3) Subsection “iPTH down-regulates signaling of elevated active TGF-β by inducing endocytosis of TβRII”: The timing is importance for changes in osteoclast numbers in subchondral bone area during OA development. The authors need to disclose this information when they describe changes in osteoclast numbers in the revised manuscript.

Thanks for the suggestions. We have added the information. “At week 4 after DMM surgery, tartrate-resistant acid phosphatase–positive (TRAP+) staining showed that the number of osteoclasts increased significantly in subchondral bone after DMM surgery; interestingly, iPTH treatment further increased the number of TRAP+ osteoclasts in subchondral bone (Figure 4G and H).”

4) Subsection “iPTH attenuates osteoarthritic pain and progression of joint degeneration”: To mimic the clinical situation, the author initiated iPTH 4 weeks after DMM surgery. The design of this study is importance since it is more relevant to the clinical situation.

Thanks for your comments and recognition.

5) In this project, the effect of iPTH was eliminated when PTH1R was deleted in Nestin-expressing cells. In addition to subchondral bone, the authors also showed the effect of iPTH in synovial tissue. The authors need to discuss external effect of iPTH besides Nestin-expressing cells.

Thanks for the suggestions. We made the relevant discussion in the revised manuscript.

Articular cartilage degeneration is the primary concern in osteoarthritis. In this study, iPTH demonstrated its protective effect on cartilage and subchondral during OA progression. Deletion of *PTH1R* in Nestin-expressing cells of PTH1R^-/-^ mice, the protective effect on cartilage remained although the effect on subchondral bone was impaired (as shown in Figure 7B), likely due to severe deterioration of subchondral bone in PTH1^-/-^ mice. Our result is consistent with previous literatures that PTH inhibit cartilage degradation, terminal differentiation, and apoptosis of chondrocytes and to promote regeneration of articular cartilage in osteoarthritis (1-4).

Regarding synovium, we did not observe significant change in PTH-treated and PBS-treated DMM mice. The synovial cells do express PTH1R, and PTH has been shown to inhibit the expression of pro-inflammatory modulators, including COX2, in the synovial membrane of the OA animal model (5). Also, the inflammatory response of synovial membrane was also reported to remain unaffected after systemic application of PTH in osteochondral defects (6). In our study, we found that the effect of PTH on relieving OA pain was remarkably blunted in the NestinCre:: PTH1R^f/f^ mice, indicating that PTH ameliorates OA pain primarily by targeting subchondral bone. The discussion has been added in the revised manuscript accordingly.

1) Bellido M, et al. Improving subchondral bone integrity reduces progression of cartilage damage in experimental osteoarthritis preceded by osteoporosis. Osteoarthritis Cartilage. 2011;19(10):1228-36.

2) Yan JY, et al. Parathyroid hormone (1-34) prevents cartilage degradation and preserves subchondral bone micro-architecture in guinea pigs with spontaneous osteoarthritis. Osteoarthritis Cartilage. 2014;22(11):1869-77.

3) Chang JK, et al. Parathyroid hormone 1-34 inhibits terminal differentiation of human articular chondrocytes and osteoarthritis progression in rats. Arthritis Rheum. 2009;60(10):3049-60.

4) Cui C, et al. Parathyroid hormone ameliorates temporomandibular joint osteoarthritic-like changes related to age. Cell Prolif. 2020;53(4):e12755.

5) Lugo L, Villalvilla A, Gómez R, et al. Effects of PTH [1-34] on synoviopathy in an experimental model of osteoarthritis preceded by osteoporosis. Osteoarthritis Cartilage. 2012;20(12):1619-1630. doi:10.1016/j.joca.2012.08.010

6) Orth P, Cucchiarini M, Zurakowski D, Menger MD, Kohn DM, Madry H. Parathyroid hormone [1-34] improves articular cartilage surface architecture and integration and subchondral bone reconstitution in osteochondral defects in vivo. Osteoarthritis Cartilage. 2013;21(4):614-624. doi:10.1016/j.joca.2013.01.008

[Editors' note: we include below the reviews that the authors received from another journal, along with the authors’ responses.]

Reviewer AGeneral Comments.This paper reports that intermittent parathyroid hormone (iPTH) attenuates osteoarthritis pain in a destabilized medial meniscus (DMM) mouse model by inhibiting subchondral sensory innervation, subchondral bone deterioration, and articular cartilage degeneration, which did not occur in mice with the PTHR conditionally KOd in Nestin+ mesenchymal stromal cells. The findings with respect to pain alleviation are novel, but they are minimal, and the statistical test used to demonstrate significant effects is inadequate, suggesting that the p values for significance for some of the data will be lower or become insignificant. In addition, some of the data confirm published effects of PTH in OA or are anticipated. Thus, although the data point to possible effects of iPTH on pain relief in OA, they do not represent a significant advance on what’s already known about PTH in this model.

Thanks for the comments. Many papers about PTH effects on OA have been published. However, they primarily investigated the effects of PTH on articular cartilage for OA treatment. OA pain is less likely derived from articular cartilage as it does not have nerve fibers and blood vessels. Instead, OA pain can be developed at a very early stage without inflammation and independent of progressive cartilage degeneration. Many asymptomatic patients have osteoarthritic radiographic changes while other patients have OA pain but with no radiographic findings. Our recent report demonstrated that subchondral bone osteoclasts induce sensory innervation for OA pain. Indeed, OA patients recerived knee replacement feel rapidly pain relief after removal of deteriorated cartilage and partial subchondral.

Pain is the most prominent symptom of OA, affecting nearly 40 million in the US alone, which may be either chronic or episodic pain. Pain is the reason people seek medical attention, and itself is a risk factor for the development of future functional decline. Increase of pain is a predictor of physical functional limitation and disability. Targeting OA pain clarified the primary effects of PTH on subchondral bone or articular cartilage. The current study in this manuscript attempts to demonstrates that PTH reduces OA pain and attenuates the progression of OA through remodeling of subchondral bone deterioration.

In addition, there were few papers about PTH on OA subchondral bone, primarily performing _µ_CT and histology analysis of subchondral bone changes. For an example, intermittent injection of PTH (iPTH) was applied in normal mice rather than OA animal models, leading to onset of OA (Orth et al., 2014). Apparently, the messages regarding the effect of PTH on OA is confusing. The objective of this study is also to provide a mechanistic insight into the effects of PTH on OA to clarify the perplexing issues. There are several novel findings regarding mechanism of the effects of PTH on OA subchondral bone:

First, iPTH stimulates new bone formation through osteoclastic bone remodeling. In normal mice with no joint OA, their subchondral bones have normal bone density and appropriate microstructure, which are in balance with articular cartilage. iPTH treatment generates additional new bone on top of normal subchondral bone density and structure. As a result, the changes of subchondral bone disrupt its balanced interplay with articular cartilage, leading to the cartilage degeneration. In OA, the subchondral bone is pathologically changed to porous structure due to uncoupled remodeling. iPTH administration stimulates osteoclast remodeling generates new bone to improve its structure quality and its interaction with articular cartilage. More importantly, the NestinCre::PTH1R^fl/fl^ mice (KO mice) were the first time created in this study. The beneficial effect of PTH on pain relief and prevention of subchondral bone deterioration were not observed in the KO mice.

In our previous study, we found that there were unmineralized or low mineralized bony tissues in the bone marrow cavity, as isolated osteoid islets in the OA subchondral bone (Zhen et al., 2013). These osteoid islets were not observed in the sham-operated control mice. In the present study, we showed that the newly formed osteoid islets in the subchondral bone marrow cavity were decreased with daily PTH injection.

Aberrant mechanical stress induces uncoupled remodeling of subchondral bone at onset of OA and subsequently generates a pathological microenvironment with significantly increased PGE2 level and other inflammatory factors when reaching a certain threshold (Rahmati et al., 2016). Importantly, subchondral bone has been shown as a source of inflammatory mediators and OA pain. The abnormal microarchitecture of subchondral bone in OA dramatically change the stress distribution on articular cartilage. iPTH modulates the mechanical stress distribution on articular cartilage by improving microarchitecture of the subchondral bone to reduce production of pro-inflammatory factors. iPTH also has been shown to ameliorate both hyperplasia and fibrosis in OA preceded by osteoporosis, and inhibit expression of pro-inflammatory modulators, including COX2, in synovial membrane (Lugo et al., 2012). In the current study, we revealed that iPTH have a therapeutic effect on OA and pain by improving subchondral bone microenvironment through remodeling, which was blunted in the NestinCre::PTH1R^f/f^ mice.

In the revised manuscript, we have substantially revised the Discussion section to clarify and highlight the significance and novelty of our study. We also improved the description of statistical analysis by adding new information. As suggested, we reanalyzed the data of

50%PWTs and PAMWTs by two-way repeated-measurement ANOVA with Bonferroni’s posthoc test. All sets of data at (2 weeks), 4 weeks, and 8 weeks were analyzed by two-way ANOVA with Bonferroni’s post-hoc test. The therapeutic effects of PTH remains significant. The detailed point to point response has been illustrated below.

Specific pointsThe title should include “in mice”.

As suggested, we have added “in mice” in the title. The current title is “Parathyroid hormone attenuates osteoarthritis pain by remodeling subchondral bone in mice”.

“For comparisons among multiple groups, we used 1-way analysis of variance.” This ideally should be followed by another test such as Dunnet’s of Bonferoni. If post-hoc analysis is done, some off the differences may become non-significant and the p values will likely decrease.

We realized that we did not describe the method for the statistical analysis clearly. We performed one-way ANOVA analysis followed by Bonferroni’s post-hoc analysis for the comparisons among multiple groups at the same time point.

We also performed additional statistical analysis. When both treatment and time course are factors (for example, the 50%PWTs and PAMWTs), variables were analyzed using two-way repeated-measures ANOVA followed by Bonferroni’s post hoc analysis. Other sets of data at (2 weeks), 4 weeks and 8 weeks were analyzed by two-way ANOVA with Bonferroni’s post-hoc test. For other comparisons among multiple groups, we used one-way ANOVA with Bonferroni’s post hoc test. These statistical analysis methods have been commonly used in the previous studies (references below). The therapeutic effects of PTH remains significant.

Ni S, et al. Sensory innervation in porous endplates by Netrin-1 from osteoclasts mediates PGE2-induced spinal hypersensitivity in mice. Nature Communications.

2019;10 (1):5643.

Wang X, et al. Aberrant TGF-_β_ activation in bone tendon insertion induces enthesopathy-like disease. J Clin Invest. 2018;128 (2):846-60.

Zhu S, et al. Subchondral bone osteoclasts induce sensory innervation and osteoarthritis pain. J Clin Invest. 2019;12z9 (3):1076-93.

Sousa-Valente J, Calvo L, Vacca V, Simeoli R, Arévalo JC, and Malcangio M. Role of TrkA signalling and mast cells in the initiation of osteoarthritis pain in the monoiodoacetate model. *Osteoarthritis and Cartilage.* 2018;26(1):84-94.

Rowe MA, et al. Reduced Osteoarthritis Severity in Aged Mice With Deletion of Macrophage Migration Inhibitory Factor. 2017;69(2):352-61.

Ishikawa G, Koya Y, Tanaka H, and Nagakura Y. Long-term analgesic effect of a single dose of anti-NGF antibody on pain during motion without notable suppression of joint edema and lesion in a rat model of osteoarthritis. *Osteoarthritis and Cartilage.* 2015;23(6):925-32.

Reference 25 is not correct.

We are sorry for the typo. The reference “Qiu T, Wu X, Zhang F, Clemens TL, Wan M, and Cao XJNcb. TGF-_β_ type II receptor phosphorylates PTH receptor to integrate bone remodelling signalling. 2010;12(3):224-34.” has been replaced with “Qiu T, Wu X, Zhang F, Clemens TL, Wan M, and Cao X. TGF-β type II receptor phosphorylates PTH receptor to integrate bone remodelling signalling. Nat Cell Biol. 2010;12(3):224-34.”.

Figure 1 A-D. Although PTH treatment improved these indices of pain and gait, the differences are small, particularly at W8, suggesting that the overall effect is minimal.

As mentioned above, we re-analyzed 50%PWTs and PAMWTs through 2-way repeated-measures ANOVA with Bonferroni’s post-hoc test. The data showed that PTH significantly improve all the indices, especially starting from 4 weeks after DMM surgery. The differences between the PTH group and vehicle group in 50%PWTs, PAMWTs, and gait analysis at 8 weeks remained significant (p< 0.01). Our results are comparable to the published studies conducting similar behavior tests (e.g. Joen et al., 2017 and Lee et al., 2018), the ratio of value of absolute difference to largest value of indices rangee from 10% to 20%.

Our results showed that the 50%PWTs of the ipsilateral limb in the PTH group is around 50% more than the vehicle treated group, starting from 5 weeks after surgery. The PAMWTs of the ipsilateral limb in PTH is over 10% more than the vehicle-treated group. Regarding gait analysis parameters, including paw intensity, contact area and swing speed, they were around 15% to 20% more than the vehicle-treated group at week 8 after DMM surgery. All the above indicates that PTH effectively reduces OA pain and improve gait in DMM mice.

Figure 2. Please state how the relative density of fibers was calculated and the area in which the measurements were made.

The intensity of positive fluorescence signals in the entire subchondral bone area was threshold (threshold range: 50-255) and calculated using Image J (version 1.49). One section at the similar sagittal location of each mouse (3 slices per mouse and 8 mice per group) was calculated and normalized to that of sham mice (set average to 1).The detailed description of the method has been added to the revised manuscript.

To illustrate the area we analyzed the fluorescence intensity of the nerve fibers, representative low magnification images of the GCRP staining in the entire subchondral bone in each group have been added to the revised manuscript (Figure 2A).

Figure 2 G-H. Please state if the +ve cells were counted in the BM or bone or both. Did PTH affect the expression in osteocytes, which appear +ve? Did PTH affect the numbers of COX-2 +ve cells in the synovium? Why did PTH have no effects in the synovium? Is PTHR expressed in synovial cells?

Yes, the COX2+ cells were counted in both bone marrow (BM) and bone (current Figure 2 K-L). The details of the quantitative analysis has been added in the corresponding figure legend. In our previous study, we have shown that decreased bone density promotes PGE2 production by osteoblasts (Chen et al., 2019), whereas osteocytes do not secrete PGE2 for low bone density. Increase of COX2 expression and elevated PGE2 levels in OA subchondral bone are associated with spontaneous OA pathological changes in mice (Tu et al., 2019). We have quantified the PGE2 levels of entire subchondral bone, and calculated the COX2 positive cells in both the BM and subchondral bone matrix in parallel. We found that PGE2 produced in both BM and bone matrix contributes to subchondral bone-related OA pain.

The synovial cells do express PTH1R, and PTH has been shown to inhibit the expression of pro-inflammatory modulators, including COX2, in the synovial membrane of the OA animal model (Lugo et al., 2012). Also, the inflammatory response of synovial membrane was also reported to remain unaffected after systemic application of PTH in osteochondral defects (Orth et al., 2013). In our study, we found that the effect of PTH on relieving OA pain was remarkably blunted in the NestinCre:: PTH1R^f/f^ mice, indicating that PTH ameliorates OA pain primarily by targeting subchondral bone. We have added this discussion in the revised manuscript accordingly.

Figure 3 A-B. These are published effects of PTH and should be in Supplementary data

Thanks for the suggestion. The changes of subchondral bone were measured by _μ_CT for the effect of PTH in the published data. In our recent study, we found that the porosity of bone induces sensory innervation and increases local PGE2 levels (Ni et al., 2019; Chen et al., 2019) to stimulate nociceptors of pain. The objective is to determine the porous structure of subchondral bone during OA progression, and whether daily injection of PTH could improve the porous structure. As shown in Figure 3A-B, subchondral bone cavities were seen in the OA mice whereas PTH treatment significantly reduced the porosity structure and reduced sensory innervation (Figure 2A-E, and Figure 3A-B). Illustration of improvement of the porosity is a necessary base to understand the elevated levels of PGE2 for the OA pain and effect of PTH on the attenuation of OA pain by blunting the noxious stimuli of PGE2 on innervated nociceptors in the subchondral bone.

Figure 3F. What are osteoid islets? This is a term with which I am unfamiliar. These and the BFR and labeling surfaces should be quantified.

In our previous study, we found that there were unmineralized or low mineralized bony tissues in the bone marrow cavity, as isolated osteoid islets in the OA subchondral bone (Zhen et al., 2013). These unmineralized or low mineralized bony tissues showed red color or very light green, which distinguished from the dark green of maturely mineralized bone in the GoldnerMasson trichrome staining. These osteoid islets were not observed in the sham-operated control mice. In the present study, we showed that the newly formed osteoid islets were decreased with daily PTH injection. The osteoids were indicated with arrowhead in the Figure 3F. As suggested, the quantification of osteoid islets in the subchondral bone cavity and BFR were also added in the Figure 3 of the revised manuscript.

Figure 4. These data largely confirm what this group has published previously in this model and represent refinement of the findings.

PTH-induced endocytosis of T_β_RII and inhibition of TGF-_β_ signaling in bone mesenchymal stem cells of WT mice was published in our previous paper (Qiu et al., 2010). However, in the current study, we attempt to investigate whether PTH-induced endocytosis of T_β_RII and inhibition of TGF-_β_ signaling improve the microarchitecture of subchondral bone in OA animal model. We found that PTH modulates TGF-_β_ signaling to restore coupled bone remodeling, and subsequently maintain the microarchitecture of subchondral bone in OA mice. As a result, PTH downregulates PGE2 production and subsequent pain signals by normalizing the microstructure of subchondral bone.

Figure 5. These effects are minimal, suggesting that PTH will likely not be an effective pain preventive medication.

As stated above, our pain behavior test results are comparable to previously published studies. After re-analyzing the 50%PWTs and PAMWTs by two-way repeated-measures ANOVA, the difference remains significant, which suggests that PTH is effective in preventing OA pain. In the experiments examining the treatment effect of PTH (Figure 5), we found PTH also attenuated OA pain even when the deterioration of cartilage and subchondral bone is already developed in mice at week 4 after DMM surgery. In detail, 50%PWTs of ipsilateral limb in PTH group was around 20% higher than the vehicle-treated group. Regarding PAMWTs, it was over 15% higher in the PTH group than that of the vehicle-treated group. The paw intensity, contact area and swing speed of ipsilateral limb in the PTH group were over 15% higher than that of the vehicle-treated group.

Figures 6-7. These results are anticipated; it would have been very surprising if the results were different.

The NestinCre::PTH1R^fl/fl^ mice (KO mice) were the first time created in this study. The KO mice provided the possibility to distinguish the effect of PTH on the subchondral bone from its effect on articular cartilage or synovium The beneficial effect of PTH on pain relief and prevention of subchondral bone deterioration not observed in the KO mice. While previously published researches primarily focus on the effect of PTH on articular cartilage, we are particularly investigating subchondral bone as a major resource for OA pain. Specifically, we found that PTH maintains the microstructure of subchondral bone by modulation of TGF-_β_ signaling in OA. The effect of PTH was abolished when PTH1R was specifically deleted in the nestin positive stromal cells. This is for the first time to investigate the mechanism of PTH in OA pain relief by normalizing the microstructure of subchondral bone and subsequent reduction of PGE2 levels.

Reviewer BThe authors aim to analyze the effects of intermittent parathyroid hormone (iPTH) treatment on osteoarthritis (OA) progression and pain sensitivity. They show that iPTH attenuates OA pain by inhibiting subchondral sensory innervation, subchondral bone deterioration, and articular cartilage degeneration in a destabilized medial meniscus (DMM) mouse model. In addition, increased level of prostaglandin E2 (PGE2) in subchondral bone of DMM mice was reduced by iPTH treatment. Effects of TGFß signalling were reduced by endocytosis of the PTH type 1 receptor–transforming growth factor β type 2 receptor (TßRII) complex. The authors conclude that iPTH could be a disease modifying OA treatment.This is a well conducted study reporting interesting data which advances the field of OA and are of interest for OA research. Study design is original and of potential clinical interest regarding novel treatment strategies of OA pathology. However, some questions remain and should be addressed.

We appreciate the reviewer’s encouraging comments.

General points- Sensory nerve fibres mostly grow or innervate tissues together with or accompanied by small blood vessels. Have the authors checked the effect of iPTH treatment on angiogenesis? This is quite important for understanding the possible mechanism by which iPHT reduces density of CGRP positive nerve fibres in the subchondral bone as the authors have not provided a reliable explanation for this specific observation. This could be done with appropriate immunostaining of existing sections.

In the revised manuscript, the co-staining of the sensory nerve (CRGP) and vessel (endomucin), and corresponding quantification have been added (Figure 2 E-F and Figure 6D-F). PTH induces osteoclastic bone resorption in coupling osteoblast bone formation during bone remodeling, and increased osteoclasts secrete PDGF-BB for type H vessel formation in support of the bone remodeling. This is a dynamic activity. Accumulatively, the daily injection of PTH for 4 to 8 weeks increases bone formation with net decrease of both total vessel formation and nerve innervation as the porous subchondral bone was decreased with new bone formation. The discussion has been added to the revised manuscript.

Sensory nerves are characterized not only by CGRP but also by Substance P (SP). In particular, SP is a pronounced peripheral occurring neuropeptide often found colocalized together with CGRP. Have you stained for SP-positive nerve fibres in response to iPTH application? This might be quite interesting in the light that iPTH may not change nerve fibre density but shift neuropeptide production from CGRP to SP.

As suggested, we performed immunofluorescence staining to label the SP positive nerve fiber, and the representative image and quantification were added in the revised manuscript (Figure 2C and D). We found that the density of SP was also reduced in the subchondral bone with iPTH treatment in OA. SP is secreted by sensory nerve fibers and share a similar distribution pattern with CRGP+ fibers. The decrease of CGRP^+^ fibers and SP in the subchondral bone with iPTH treatment suggests SP is largely secreted from CGRP^+^ fibers.

A highlight of OA is osteophyte formation also in the DMM model. Osteophytes can be responsible for increased pain sensation and abnormal gait behaviour. Have you detected osteophytes and if yes, is there any effect of iPTH treatment on osteophyte formation or osteophyte structure?

We appreciate your suggestion. We performed additional _μ_CT analysis specifically examining the size of osteophyte in OA mice. We did not observe significant difference in osteophyte formation between iPTH and vehicle in DMM mice, in consistence with published data “treatment with Zol and PTH did not influence the size of the osteophyte formation” in the OA model (Bagi et al., 2015). The quantification of volume of osteophyte in each group were add in the revised manuscript (Figure 2I-J, 6I-J).

Specific points ResultsWhy is there no overlap of P2X3, PIEZO2 and CGRP positive neurons (Figure 2A)?

Likely, they were not good sequential sections. In the revised manuscript, we performed co-immunostaining of P2X3 with CGRP, and PIEZO2 with CGRP for nociceptors in the subchondral bone. The results showed CGRP+ nerve fibers partially overlap with P2X3+ or PIEZO2+ nerve fibers (Supplementary Figure 1A-B).

It is not possible to decide for the reader whether the osteoids are located at the bone surface or not (Figure 3F). How did you specify this location?

Yes, osteoids are normally located at the bone surface in WT mice. When osteoids are not on the bone surface as “osteoids islets”, indicating the uncoupled formation of pathological changes. In the revised manuscript, we performed sequential sections to confirm the isolation and provide quantification of “osteoid islets” in the subchondral bone cavity (Figure 3 F-H).

DiscussionIt should be discussed why the density of CGRP positive fibres in the synovium remains unchanged after PTH treatment whereas in the subchondral bone their density decreases.

We appreciate your suggestion. We have revised the Discussion section and added new information regarding the change of sensory nerve in the synovium and the subchondral bone in the revised manuscript.

In general, PTH attenuates OA by both maintaining microstructure of the subchondral bone and the protection of articular cartilage. As shown in our study, PTH effectively reduce density sensory nerve and vessel in the subchondral bone in DMM mice. The effect of PTH on OA pain is less due to direct protection of cartilage because cartilage is aneural and lack of vessel.

Furthermore, when systemic application of PTH was used for the repair of nonosteoarthritic, focal osteochondral defects in vivo, synovial membrane remained unaffected (Orth et al., 2013). In our study, we do not find a significant difference in nerve density between PTH and vehicle group (Figure 2 G and H). This finding indicates PTH does not have a direct effect on nociceptive innervation in the synovium.

It should be also discussed why reduction of sensory nerve fibers in the subchondral bone only has a pain reducing effect. What is the possible contribution of the unchanged density of sensory nerve fibres in the synovium to the pain sensation?

It has been reported that sensory innervation promotes osteoblast bone formation and suppresses osteoclast bone resorption (Li et al., 2017 and Ishizuka et al., 2005). We have shown that sensory nerve regulates bone homeostasis and promote regeneration (Chen et al., 2019). The ablation of sensory innervation by genetic or pharmacological approaches consistently results in decreased bone mass in adult mice (Chen et al., 2019; Brazill et al., 2019 and Fukuda et al., 2013).

As discussed above, PTH induces osteoclastic bone resorption in coupling osteoblast bone formation during bone remodeling. Increased osteoclasts secrete PDGF-BB for type H vessel formation in support of the bone remodeling. This is a dynamic activity. Accumulatively, the daily injection of PTH for 4 to 8 weeks increases bone formation with net decrease of both total vessel formation and nerve innervation as the porous subchondral bone was decreased with new bone formation. Therefore, iPTH decreased sensory nerve and attenuated OA pain by a decrease of porous subchondral bone.

Sensory nerve in the synovium likely also contributes to pain in OA. However, we did not observe a significant effect of PTH on the nociceptive innervation in the synovium membrane. Experiments of Nestin Cre-PTH1R f/f mice suggest the beneficial effect of PTH primarily through its effect on subchondral bone.

As suggested, the discussion has been added to the revised manuscript.

Materials and methodsMice and in vivo treatmentPlease specify nature of PTH and include information about origin. Either provide ratio for the PTH dose and mode of injection you have used or include reference. Please indicate PTH treatment starting time point.

As suggested, the nature and information about the origin of PTH were specified. The information of PTH dose, mode of injection and the starting time point of PTH treatment were also added in the Materials and methods section.

Briefly, sham mice or DMM mice were injected subcutaneously with 40 µg·kg-1 per day of human PTH (1-34) (Σ-Aldrich, USA) or the equivalent volume of vehicle (phosphatebuffered saline [PBS]). For the immediate regime of administration, the daily injection of PTH was initiated 3 days after DMM surgery. For delayed regime of administration, the daily injection of PTH was initiated 4 weeks after DMM surgery.

N=8 is a low group size for behavioural tests. Usually, larger group sizes are used, i.e.N=15. Please comment on this.

The same behavior tests were frequently used in our previous studies. Our results showed that 8 mice per group in these behavioral tests usually can achieve statistical significance (Zhu et al., 2019 and Ni et al., 2019). Some other research group also used 8-12 mice per group in their behavioral test (Jeon et al., 2017). Previous literature used 50%PWTs to evaluate OA pain, they also state “For behavioural analysis, based on our previous experience and data, in order to achieve α 0.05 and power 0.8 we used at least six animals per group” in the data analysis section (Souse-Valente et al., 2018). In the present study, the DMM surgery was performed by investigator with skillful technique, thus the variation within a group is relatively small. The treatment effect of PTH is dramatic enough to show statistical difference when compared to the vehicle-treated group using 8-12 mice per group.

Moreover, the average of three measurements was taken for each time point to eliminate potential bias in our study.

It is recommended by the OARSI to use 12 weeks old mice for DMM induced OA. Please comment on why you have used 10 weeks old mice instead.

Both 10 weeks and 12 weeks mice are used in our lab to develop the DMM animal model. In our experience, no apparent difference was observed between 12 weeks and 10 weeks old mice after DMM surgery when grading the cartilage degeneration using the OARSI score system. Additionally, 10-12-week-old male C57BL/6 mice were frequently used and qualified in the DMM model by other research groups. The representative publications are listed below.

Kung LHW, et al. Cartilage endoplasmic reticulum stress may influence the onset but not the progression of experimental osteoarthritis. Arthritis research & therapy. 2019;21(1):206-.

Li J, et al. Metformin limits osteoarthritis development and progression through activation of AMPK signalling. Annals of the rheumatic diseases. 2020;79(5):635-45.

Hong JI, Park I, Kim JR, and Kim H. Study on the gender dimorphism and TRPV1 expression on chronic pain in DMM-induced osteoarthritis model mice. Osteoarthritis and Cartilage. 2019;27:S417.

Li H, et al. Exploration of metformin as novel therapy for osteoarthritis: preventing cartilage degeneration and reducing pain behavior. Arthritis Research & Therapy. 2020;22(1):34.

Histochemistry, IHC and HistomorphometryPlease describe in more detail how you have performed counting of nerve fibers as this is not trivial. You would need some sort of metric system here.

The intensity of positive fluorescence signals in the entire subchondral bone area was threshold (50-255) and calculated using Image J (version: 14.9). One section at the similar sagittal location each mouse was calculated and normalized to that of sham mice (set average to 1). As suggested, a detailed description of the method has been added to the revised manuscript.

µCTWere the same samples first used for µCT and afterwards for histology however that depends on scan time? How long was the average scan time of the samples?

Yes, the same samples were used for µCT and followed by histology analysis. The average scan time of the knee joint was 20-25 min (The scanner was set at a voltage of 65 kV, a current of 153 _μ_A, and a resolution of 9 _μ_m per pixel). Before the µCT scan, the samples were fixed in 10% buffered formalin for 48 h. After CT scan, the samples were subjected to decalcification in 10% EDTA for 2 weeks. In our hand, there is no difference in histological analysis outcome between samples with or without a CT scan beforehand.

ELISADescribe the sample preparation for the PGE2 ELISA in detail, as the manufacturer does not include description for preparation of hard tissues as bone for this ELISA.

As suggested, in the revised manuscript, we have made a detailed description of the sample preparation for the PGE2 ELISA in the Materials and methods section.

Briefly, we snap-freezed and crushed the subchondral bone in liquid nitrogen, right after tissue harvest and repaid removal of surrounding tissue and cartilage. Then homogenization buffer (0.1M phosphate, pH7.4, containing 1mMEDTA and 1_0μ_M indomethacin) were added (ratio: 5 ml buffer to 1g tissue). The sample were homogenized with a Polytron-type homogenizer. The tissue homogenates were spun at 8000xg for 10 minutes. The supernatant was collected for ELISA assay, according to the manufacturer's instructions.

FiguresGeneral: The paper would benefit from a figure, which summarizes the main findings of this study.

We appreciate your suggestion. A schematic diagram summarizing the main findings has been added in the revised manuscript as Figure 8.

Figure 1Please explain meaning of #? To which group does this symbol relate (A,B)?

Sorry for the missing information. The symbol of “#” refers to that vehicle group versus the sham group in both panels A and B. The symbol of “*” refers to PTH group versus the vehicle group in both panels A and B. We have added the definition of the symbols in the revised figure legends.

Please indicate age of mice used for sections shown under E and G.

As suggested, the age of mice (18 weeks old) for panel E and G were added in the revised figure legends accordingly. Additionally, we have added the age of mice used in each figure legend.

Figure 2Please outline shape of tibia representatively (A). Otherwise, it is difficult to relate the location of the nerve fibres to the joint.

We appreciate your suggestion. Low magnification images of the GCRP staining in the entire subchondral bone have been added in the Figure 2A to more clearly illustrate the area and morphology of the tibia subchondral bone. The contour of the tibial subchondral bone has been outlined.

Label the images under G appropriately. Which part of the tibia is shown here? Again, please indicate age of mice used for sections. This goes for all further figures!

As suggested, the positive staining of COX2 was labeled with arrow head in panel L. The staining of COX2 was done in the sagittal sections of medial compartment of the mouse tabial subchondral bone. We have added the information regarding the age of the mice in all figure legend including Figure 2K (original Figure 2G).

Reviewer COsteoarthritic pain is a major unattended medical problem that greatly reduces quality of life in millions of people worldwide. The study is interesting. The experiments with conditional deletion of PTH1R are particularly powerful.

We appreciate the reviewer’s encouraging comments.

Statistical method. The methods used for analysis and their description needs improvements. The type of post-hoc tests used for analysis should be stated in the Materials and methods. The methods and the figure legend should specify whether the data were normally distributed. Data in Figure 1A, Figure 1B, Figure 5A and Figure 5B should be analyzed using ANOVA for repeated measures and appropriate post-hoc tests. In these figure the symbols “*” and “#” are not defined. It is unclear which points are significantly different as compared to baseline and the corresponding time point of other groups.All sets of data obtained at 4 weeks and 8 weeks should be analyzed by two-way-ANOVA as opposed to one-way-ANOVA. It is important to show that statistically significant effects of PTH remain so when data are analyzed by two-way-ANOVA and post-hoc tests.

We apologize that the method used in the statistical analysis were not described clearly. In the revised manuscript, we have added a detailed description of analysis. We performed one-way ANOVA analysis followed by Bonferroni’s post-hoc analysis for the comparisons among multiple groups. One-way ANOVA followed by Turkey’s or Bonferoni’s post hoc analysis is a routine analysis among multiple groups (Ni et al., 2019; Wang et al., 2018 and Zhu et al., 2019). The status of data distribution is also described in Materials and methods. The data were normally distributed, unless otherwise noted.

We have added definition of “*” and “#” in the figure legend of revised manuscript.

As suggested, we performed 2-way repeated-measures ANOVA with Bonferroni’s post-hoc test to evaluate the effect of PTH on PAMWTs and 50%PWTs at different time points. Other sets of data at (2 weeks), 4 weeks and 8 weeks were analyzed by two-way ANOVA with Bonferroni’s post-hoc test. For other comparisons among multiple groups, we used one-way ANOVA with Bonferroni’s post hoc test. These statistical analysis methods have been commonly used in publications (Ni et al., 2019; Wang et al., 2018; Zhu et al., 2019; Souse-Valente et al., 2018; Rowe et al., 2017 and Ishikawa et al., 2015).

Figure 3. In several panels, the y-axis does not start at 0, increasing the appearance of large differences among groups. Please edit the figure and set the origin of the y-axis at 0.

As suggested, we have made changes for the figures and set the y-axis started at 0 in the revised manuscript.

Discussion: This section could be improved by addressing in more depth the fact that PTH has been previously reported to aggravate osteoarthritis, while in the current study PTH was found to have beneficial effect. How is this contradiction explained? Similarly, osteoarthritis is currently regarded as an inflammatory disorder. Do the findings of the current study support the inflammatory theory? If not, how are the mechanistic differences explained?

Thanks for the constructive suggestions. Yes, iPTH has been reported to trigger the onset of OA in animal models, in which iPTH was administrated in the normal mice rather than OA animal models (Orth et al., 2014). iPTH stimulates new bone formation through osteoclastic bone remodeling. In normal mice with no joint OA, their subchondral bones have normal bone density and appropriate microstructure, which are in balance with articular cartilage. iPTH treatment generates additional new bone on top of normal subchondral bone density and structure in mice without OA. As a result, the changes of subchondral bone disrupt its balanced interplay with articular cartilage and leads to degeneration. In OA, the subchondral bone is pathologically changed to a porotic structure due to uncoupled remodeling. iPTH administration stimulates osteoclast remodeling generates new bone to improve its structure quality and its interaction with articular cartilage.

Aberrant mechanical stress induces uncoupled remodeling in the subchondral bone at onset of OA and subsequently generates a pathological microenvironment with significantly increased PGE2 level and other inflammatory factors when reaching a certain threshold (Rahmati et al., 2016). Importantly, subchondral bone has been shown as a source of inflammatory mediators and OA pain. The abnormal microarchitecture of subchondral bone in OA dramatically change the stress distribution on articular cartilage. Also, iPTH modulates the mechanical stress distribution on articular cartilage by improving microarchitecture of the subchondral bone to reduce production of pro-inflammatory factors. iPTH also has been shown to ameliorate both hyperplasia and fibrosis in osteoarthritis preceded by osteoporosis, and inhibit expression of pro-inflammatory modulators, including COX2, in synovial membrane (Luo et al., 2012). In the current study, we reveal that the therapeutic effect of iPTH on OA and pain by improving subchondral bone microenvironment through remodeling, which was blunted in the NestinCre::PTH1R^f/f^ mice.

The Discussion and related references have been added in the revised manuscript.